

# Using 3D observations with high spatio-temporal resolution to calibrate and evaluate a process-focused cellular automaton model of soil erosion by water

Anette Eltner[1], David Favis-Mortlock[2], Oliver Grothum[1], Martin Neumann[3], Tomáš Laburda[3],
Petr Kavka[3]

[1] Institute of Photogrammetry and Remote Sensing, TUD Dresden University of Technology, Dresden, Germany
[2] British Geological Survey, Nicker Hill, Keyworth, Nottingham, NG12 5GG, UK
[3] Faculty of Civil Engineering, Czech Technical University in Prague, Prague, Czechia

*Correspondence to*: Anette Eltner (anette.eltner@tu-dresden.de)

**Abstract**. Future global change is likely to give rise to novel combinations of the factors which enhance or inhibit soil erosion by water. Thus there is a need for erosion models, necessarily process-focused, which are able to reliably represent rates and extents of soil erosion under unprecedented circumstances. The process-focused cellular automaton erosion model RillGrow is, given initial soil surface microtopography on a plot-sized area, able to predict the emergent patterns produced by runoff and erosion. This study explores the use of Structure-from-
Motion photogrammetry as a means to calibrate and validate this model by capturing detailed, time-lapsed data for soil surface height changes during erosion events. Temporally high-resolution monitoring capabilities (i.e. 3D models of elevation change at 0.1 Hz frequency) permit validation of erosion models in terms of the sequence of formation of erosional features. Here, multi-objective functions, using three different spatio-temporal averaging approaches, are assessed for their suitability in calibrating and evaluating the model's output. We used two sets of
data, from field- and laboratory-based rainfall simulation experiments lasting 90 and 30 minutes, respectively. By integrating 10 different calibration metrics, the output of 2000 and 2400 RillGrow runs for the field and laboratory experiments respectively, were analysed. No single model run was able to adequately replicate all aspects of either field and laboratory experiments. The multi-objective approaches highlight different aspects of model performance, indicating that no single objective function can capture the full complexity of erosion processes. They also
highlight different strengths and weaknesses of the model. Depending on the focus of the evaluation, an ensemble of objective functions may not always be necessary. These results underscore the need for more nuanced evaluation of erosion models, e.g. by incorporating spatial pattern comparison techniques to provide a deeper understanding of the model's capabilities. Such evaluations are an essential complement to the development of erosion models which are able to forecast the impacts of future global change.

**1 Introduction**

Soil erosion by water is an environmental problem of global significance (e.g. Nearing et al., 2017; Quinton and Fiener, 2023). In the future, it is likely to become more pressing in locations where anthropogenically-driven climate change brings more, and/or more intense, rainfall; and/or where changes in land usage (resulting from changes in climate, economic factors, and/or other drivers) operate to leave soil unprotected by vegetation at times
of heavy rainfall (e.g. Boardman et al., 1990; Favis-Mortlock and Boardman, 1995; Li and Fang, 2016; Dunkerley,



2019, Chen et al., 2024, Zhao et al., 2024). Such future changes are likely to result in novel combinations of the factors which cause or inhibit erosion (Foucher et al., 2024).

To manage soil erosion by water, quantification of the rate and extent of erosion is essential. Modelling is a primary tool for such quantification. However, when aiming to model erosion under novel circumstances, it is unwise to

make use of models which work in a wholly 'black box' manner, that is by extrapolating from previously encountered combinations of erosion causing/inhibiting factors. Such models (e.g. Wischmeier, 1976; Renard et al., 1991; Panagos et al., 2015) cannot represent the impacts upon erosion rates and extents due to currently unknown thresholds in, or non-linearities of response to, the erosion causing/inhibiting factors. It is these thresholds and non-linearities which will provide the greatest surprises with regard to future erosion. Thus, there is a vital scientific

need to improve, and to continue to improve, our understanding of the processes of soil erosion by water, and to incorporate this understanding in quantitative process-focused models; with (at the same time) such models ideally making use of readily-available data sources. This is a considerable challenge: but only by doing this will we be able to satisfactorily manage future soil erosion by water.

While there are many process-focused models which simulate the effects of soil erosion by water (e.g. Jetten et

al., 1999; Battista et al., 2019; Raza et al., 2021; Rose and Hadaddchi, 2023), the RillGrow model (Favis-Mortlock, 1998) is unusual in adopting a cellular automaton (CA: see list of acronyms below) representation of the eroding soil surface (cf. Smith, 1991; Murray and Paola, 1997; Coulthard et al., 2002; Darboux et al., 2002; Nicholas, 2005). CA models have been used to study emergent phenomena in a wide variety of scientific domains (e.g. Wolfram, 1982; Wu, 1998; Cappuccio et al., 2001; Wahle et al., 2001; Silva et al., 2019; Favis-Mortlock, 2004).

In RillGrow – as in the majority of CA models – all process interactions are 'local', i.e. take place only between adjacent cells of the DEM grid which represents soil surface elevations (and other soil properties). There are no process representations which operate on the DEM as a whole. As a consequence of this purely local focus, the model makes no distinction between rill and inter-rill erosion processes. Instead, they are considered to be part of a continuum. The model's local (i.e. confined to a single cell and the cells which surround it) representation of

erosion processes creates larger emergent multi-cell patterns: microrills and rills (Favis-Mortlock et al., 2000).

In essence, RillGrow works as follows. All or part of rain which falls onto the DEM becomes runoff. This moves down the steepest D8 adjacent cell-to-cell microtopographic slope, removing soil as it flows (this is the FD-FT erosional subprocess in Kinnell, 2001). Runoff continues moving downslope until it either leaves the DEM, or accumulates as a pond in a microtopographic hollow. Ponds deepen until they overtop. Runoff leaving ponded

areas continues, as previously, down the steepest adjacent cell-to-cell slope. In this way microrills are incised; their location having been determined only by microtopography. Microrills 'compete', with the most 'successful' (i.e. those that have been incised in such a way that they become part of a connected network which conveys runoff downslope) growing further to become rills. Eventually, a highly-connected rill network is formed by the model. The model uses a timestep for each iteration which is dynamically controlled by the maximum speed of cell-to-

cell runoff during the previous timestep, to ensure that cell-to-cell flow in the model obeys the Courant condition (Courant et al., 1928). With cell sizes of mm to cm, timesteps are of the order of hundredths of a second. Thus, RillGrow simulations require considerable computing power for large grid sizes: simulations are therefore (with current versions of the model) confined to plot-sized areas. In addition to erosion by overland flow, recent versions



of RillGrow also consider deposition, soil redistribution by rainsplash (RD-ST and RD-FT: Kinnell, 2001), grav-
itational collapse, and infiltration. The model is described in more detail in section 2.3.

As with virtually all erosion models (e.g. Favis-Mortlock et al., 2001), there is a need to calibrate the empirical
inputs which RillGrow requires, and then to evaluate model results against observations (Jetten et al., 1999; Batista
et al., 2019). The relatively unusual modelling approach adopted by RillGrow lends itself to the exploitation of
novel tools for calibration and validation (Epple et al., 2022). This is particularly so with regard to the capture of
spatial patterns, such as rill networks.

In an early RillGrow study (Favis-Mortlock, 1998), a moving-head laser scanner was used to capture microtopog-
raphy: first of the initial soil surface of a laboratory-based plot, and then of the eroded soil surface of the plot
following simulated rainfall. The initial-surface DEMs which resulted from these scans were used as input to
RillGrow, and the end-of-experiment DEMs were used to validate the model's output. This strategy – comparing
the model's spatial output only with the end-of-experiment DEM – however leaves open the possibility of "the
right answer for the wrong reason" since modelled rills may form in a temporal sequence which is different from
reality. The sequence with which erosional channels are incised (i.e. the dynamic development of flow routing
patterns) is of major importance when considering temporal changes in connectivity on areas ranging from plot-
sized to field-sized (Baartman et al., 2020).

Thus, there is a need for intra-experiment DEM captures ("time slices") to improve model validation. A subsequent
laboratory-based RillGrow study used data from Helming et al. (1998) and did make use of intra-experiment
DEMs. However, it was necessary to pause the simulated rainfall in order to use the laser scanner to capture the
intra-experiment microtopography, then to restart the simulated rainfall. These within-experiment pauses were
necessary for two reasons. Firstly, because laser scanning could not be carried out with simulated rain falling on
the moving scanner head (and even if it could, the scanner head would interfere with the uniformity of the simu-
lated rain). Secondly, because laser scanning as used in this experiment was not instantaneous: the scanner required
some minutes to cover the whole plot area. Pausing an experiment in this way, to capture a snapshot of rapidly
changing microtopography, is potentially problematic. Diminishing flow in rills during intra-experiment stoppages
will result in within-rill deposition, which may influence subsequent within-rill detachment when rainfall is re-
started; also, a newly-developed soil crust (or "seal") may begin to dry out and so change its properties, particularly
with regard to infiltration.

Another RillGrow study (Favis-Mortlock et al., 1998) made use of simple photogrammetry to capture initial and
final DEMs of a small area, together with a single intra-experiment DEM which was obtained without pausing the
experiment. Whilst this was a step forward, a single intra-experiment DEM is not enough to satisfactorily validate
the temporal sequence with which erosional patterns form; also, there were problems with raindrops obscuring
parts of the eroding area.

SfM photogrammetry can easily capture multiple intra-experiment DEMs because almost all processing is auto-
matic (Eltner and Sofia, 2020), it therefore promises to be a useful tool to calibrate and validate spatially-explicit
erosion models such as RillGrow. SfM photogrammetry permits intra-experiment measurement of changes of soil
surface microtopography at mm to cm resolution on plot- and hillslope-sized areas, with mm to cm accuracy (e.g.
Eltner et al. 2018, Hänsel et al., 2016). So far, SfM photogrammetry in the field of erosion studies has mainly been
applied to UAV imagery. However, its potential for application with terrestrially-installed camera systems to



increase the frequency of geomorphic change detection to hours (e.g. Blanch et al., 2024) or even to seconds (Eltner et al., 2017) has been illustrated. Note that with time-lapse SfM photogrammetry, falling raindrops obscur-

ing the plot or hillslope are not usually an issue (unlike the approach used in Favis-Mortlock et al., 1998) due to the high frequency of data capture.

The aims of this study were first to use several multi-objective functions to calibrate and evaluate a process-focused soil erosion model (RillGrow), and then to evaluate these objective functions in terms of information gained from each function. We used time-lapse SfM photogrammetry and measured sediment yield from two plot-sized rainfall

simulations, one in the field and the other in the laboratory. Ten different objective functions were considered to find the best model parameters. The best-performing model runs were chosen to be those with the lowest residuals for multiple objective functions.

## 2 Methods

### 2.1 Data acquisition

The field and laboratory rainfall simulators of the Czech Technical University in Prague were used in this study (Figure 1). Rainfall simulators are utilized to monitor surface runoff and erosion processes because they can help to accelerate erosion processes for a faster monitoring of land surface changes. Simulators have become an integral part of erosion research and erosion model calibration (Iserloh et al., 2013, Prosdocimi et al., 2017, Bosio et al., 2023).

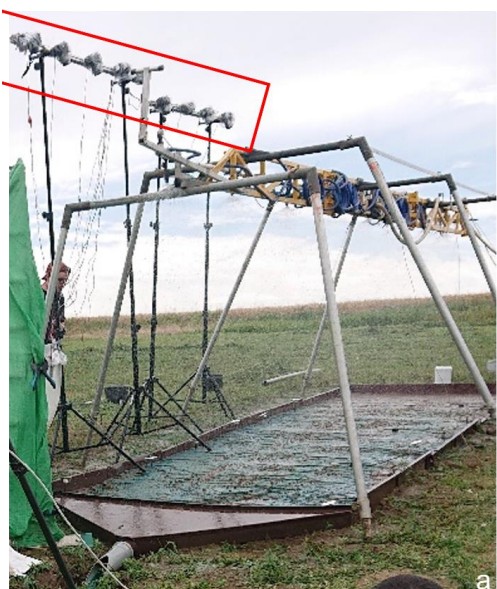
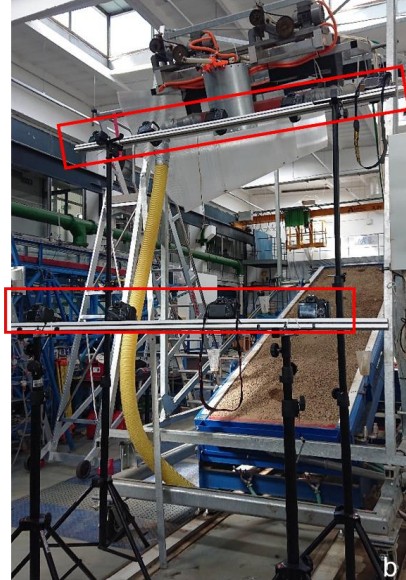

**Figure 1: Rainfall simulator used in the field (a) and in the laboratory (b).**



During the field experiment, the rainfall simulator (Kavka, 2018) was used on a plot that had been prepared as cultivated fallow. In the laboratory, the rainfall simulator (Kavka, 2019) was used on a disturbed soil sample which had been prepared similarly to the field experiment. Further site and experiment properties are given in Table 1.

Both rainfall simulation experiments used a similar approach regarding runoff and sediment sampling according

to a standard procedure for conducting experiments with the CTU rainfall simulators (Stašek et al., 2023). On the lowest side of the plot surface runoff is concentrated by a metal funnel and samples are collected into sampling vessel with volumes of one to two litres every 2.5-minute after runoff started. The samples were weighted to obtain the volume of surface runoff. Then, they were filtered by paper filtres KA-3M (Papírny Pernštejn, Czechia) and dried at 105°C in an air drier to obtain the amount of soil per sample.


**Table 1: Rainfall, plot and soil characteristics for both field and laboratory rainfall simulation experiments**

|  | Field experiment | Laboratory experiment |
|---|---|---|
| *Rainfall properties* |  |  |
| Rainfall intensity (mm h$^{-1}$) | ~140 | ~120 |
| Kinetic energy (J m$^{-2}$ mm$^{-1}$) | 10 | 7.8 |
| Nozzle type | WSQ40 | WSQ40 |
|  |  |  |
| *Plot properties* |  |  |
| Rained-on area (m) | 2 x 8 | 1 x 4 |
| Slope (%) | 9 | 40 |
| Duration of experiment (min) | 90 | 30 |
| Surface | Seed bed condition (cultivated fallow/bare soil) | |
|  |  |  |
| *Soil properties* |  |  |
| Soil type | Loam | |
| Clay/Silt/Sand % | 10.5 / 56.6 / 32.9 | |
| Organic carbon % | 1.49 | |
| Soil depth and condition | Undisturbed soil - arable soil with 35 cm deep topsoil, cultivated to the depth of 10 cm with a hand cultivator and hand roller (Stašek et al., 2023) | Disturbed soil - 15 cm "topsoil" and 5 cm of sand "bottom". After filling rolled |

Nine SLR cameras (Canon EOS 450D, 600D, 1100D, 2000D and Nikon D700) were used to create the photogrammetric data for both field and laboratory experiments (Figure 2). These captured images every ten seconds in

a synchronized manner using a remote trigger which had been constructed in-house. The cameras used different, but fixed, focal lengths to ensure a stable principal distance during the experiment. At the field site the cameras were mounted at a height of about 4 m. This captured a region of interest (RoI) of about 4 m² covering the lower part of the plot. In the laboratory the whole plot was captured (about 4 m²). In addition, images were captured with a Sony Alpha 6600 (142 and 148 images at the field and 77 and 79 images at the laboratory site before and after

the experiment, respectively), by walking around the plot to measure the entire area underneath the rainfall simulator. Ground control points (GCPs) were distributed around the plot (15 and 22 for the field and laboratory experiments, respectively) and measured with mm-accuracy using a total station, in order to scale and reference the image measurements.



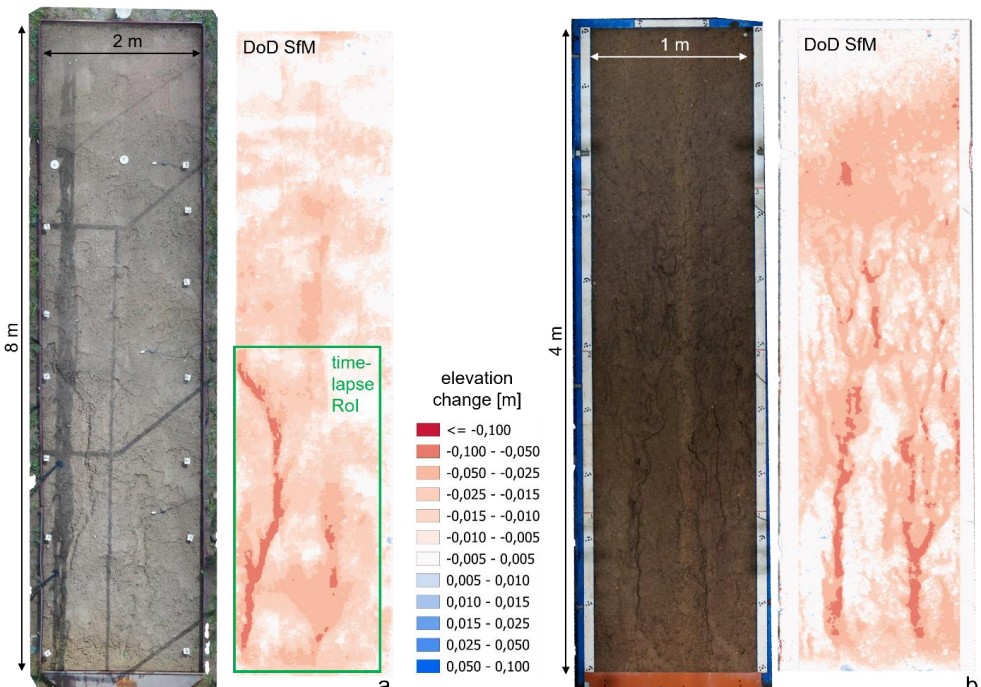

**Figure 2: Ortho photo (left) and elevation change map (right) of the field (a) and laboratory (b) experiments. The cameras for time-lapse SfM are marked with red boxes. The green box (in a) shows the region of interest, for which temporal high resolution change observations were made.**

## 2.2 Image data processing

Images that were captured with the Sony Alpha 6600 were used to derive 3D representations of the whole plot for both field and laboratory experiments with the highest possible quality and resolution, using Agisoft Metashape (v. 2.0.2) and following the standard SfM photogrammetry steps (e.g. Eltner and Sofia, 2020). Processing of the time-lapse image data involved some more steps to get the 3D models and was done according to Grothum et al.

(2024). Images were sorted according to their acquisition time and then processed using the API of Metashape (v. 2.0.2) to automatically generate a time series of DEMs using time-lapse SfM photogrammetry (Eltner et al., 2017, Blanch et al., 2024). In total, 708 and 220 filtered (for outliers), dense 3D point clouds were calculated for the field and laboratory experiments, respectively. The point clouds from the whole plot and time series were then rasterized using an interpolation approach that retains the average height value for points falling in the same raster cell.

Empty cells were linearly interpolated considering the nearest non-empty cells. For both field and laboratory experiments, the image-based 3D models of the whole plots were rasterized to a resolution of 3 cm and 1.5 cm, respectively. These DEMs were then used as input to the RillGrow model. The time-lapse data were rasterized to resolutions of 1 cm and 0.5 cm for field and laboratory, respectively. Note that the time series of DEMs covers a smaller RoI at the field site because the cameras were not able to cover the whole erosion plot.



DoDs were also constructed by point cloud differencing considering M3C2-PM (James et al., 2017) to estimate significant changes based on the accuracy of the image-based 3D reconstruction. In this way, the variance of the tie points, resulting from the bundle adjustment, is applied to estimate the spatially distributed error, which is then transferred to the dense point cloud using a distance-based weighted average if several points fall within a given search radius. Considering error propagation, the information is then used with the multi-scale cloud-to-cloud

approach (M3C2, Lague et al., 2013) to calculate the point cloud differences. The first point cloud of the time-series is used as a reference point cloud to which all subsequent point clouds are differentiated, to ensure the same orientation of the point normal used by the M3C2 tool. The final point clouds of difference are rasterized to DoDs with the same resolution as the time-lapse DEMs (i.e. 1 cm and 0.5 cm) considering only the significant changes and using the M3C2 distance as the Z (vertical) value. However, no interpolation was performed this time due to

large data gaps especially at the beginning of the experiments, when there were only small changes in soil surface elevation which fell within the noise level of the data.

### 2.3  The simulation model for soil erosion by water

Early versions of RillGrow (Favis-Mortlock, 1998) adopted simple, mostly empirical, representations of the erosion and deposition due to overland flow. Nonetheless, the model was able to satisfactorily replicate spatial patterns

of observed rill networks and amounts of runoff and soil loss on plots in the laboratory and field (Favis-Mortlock et al, 1998; 2000). Subsequent development of the model has moved towards representations which are more – but still, unavoidably, not wholly – physics-based. A flowchart of the model, outlining its representation of hydrological and erosional processes, is shown in Figure 3.

Rain falls onto the grid at random locations, as individual drops. The number of drops per timestep is calculated

from the mean and standard deviation of rainfall intensity, both of which are user inputs. Raindrop volume is calculated from the mean and standard deviation of raindrop diameter, which are use inputs. Raindrop fall velocity is also a user input.

Infiltration (if chosen by the user; otherwise, the soil is assumed to be saturated) is calculated using the explicit Green-Ampt Model of Salvucci and Entekhabi (1994). Overland flow moves from grid cell to grid cell in the D8

direction of steepest slope until it leaves the grid or can move no further. The depth of water in ponded cells increases as more overland flow arrives at the cell. Eventually overtopping may occur. The speed of overland flow on the grid is calculated using either a Manning-type or the Darcy-Weisbach flow velocity equation: if this is chosen then the friction factor may be a user-input constant, based on the Reynolds' number, or calculated using the Lawrence (1997) approach. No distinction is made between rill and inter-rill overland flow. In the version of

RillGrow used in this study, splash redistribution was calculated using the diffusion equation approach of Planchon et al. (2000), together with a water depth-splash efficiency relationship. This represents RD-ST in the Kinnell (2001) classification of erosion subprocesses.

On each wet cell, transport capacity is calculated using equation 5 from Nearing et al. (1997). If a wet cell's sediment load is less than the transport capacity, then soil is eroded from the cell using a probabilistic detachment

equation by Nearing (1991). This represents FD-FT in the Kinnell (2001) classification of erosion subprocesses.



Note that no distinction is made between rill and inter-rill flow erosion. If a wet cell's sediment load is greater than the transport capacity, then soil is deposited on the cell using equation 9 from Cheng (1997).

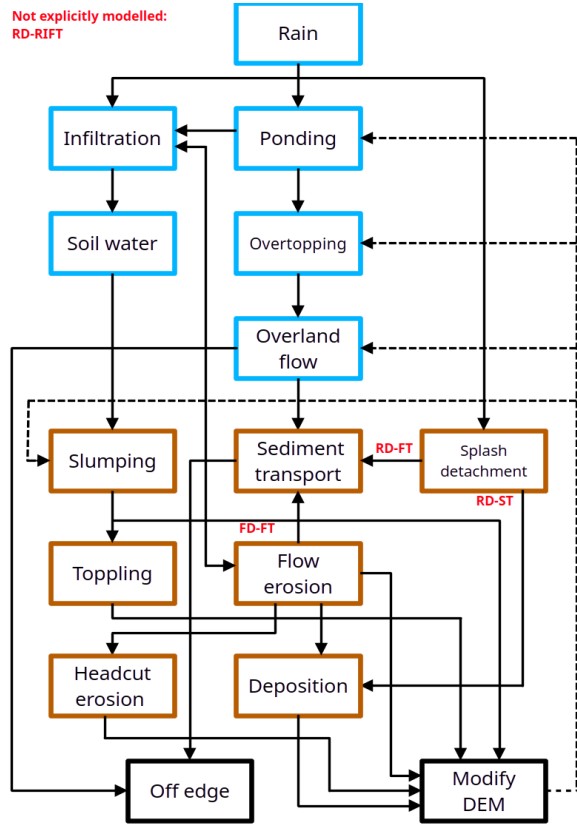

**Figure 3: Flow chart of RillGrow. Blue-edged boxes indicate hydrological processes or stores, brown-edged boxes indication erosional processes, black-edged boxes are model specific. The codes (RD-ST, FD-FT etc.) describe erosional subprocesses, see Kinnell (2001). Plain black lines represent process linkages, dashed black lines represent feedback.**

In the RillGrow model "slumping" represents mass movement on wet cells: it occurs most frequently along rill sidewalls. If a cell's shear stress (due to overland flow) exceeds a user-input threshold, and the cell is saturated (i.e. subsurface soil water for the cell is at its maximum value) then slumping occurs. Soil is assumed to flow hydrostatically down the steepest D8 soil-surface gradient surrounding the cell until it reaches a user-input angle of rest. "Toppling" similarly represents mass movement, but the soil cell does not need to be saturated: if any cell-cell gradient exceeds a user-input threshold, then soil is assumed to move until it reaches a user-input angle of rest. A representation of headcut erosion is still under development: in the version of the model used in this study, if the cell-cell gradient at the extreme upslope end of a rill exceeds an empirical threshold, headcut erosion occurs, resulting in upstream movement of the headcut. Since headcut erosion is relatively slow, it is represented using



'stored retreat' in increments which are less than the size of a cell, until the 'stored retreat' is equal to the cell size. At this point, the cell is eroded and incremental headcut erosion begins in the next cell upstream.

Detached and deposited sediment is considered to comprise three size fractions (clay, silt, sand). The soil grid is represented as one or more erodible layers above an unerodible basement. Each soil layer can possess different erodibilities for flow, splash, and slumping, for each of the three size fractions.

### 2.4 Input parameters and model runs

To perform the erosion modelling with a range of model input parameters, a Monte Carlo-like approach was used.
This employed Latin hypercube (LHC) sampling to draw the parameters approximately randomly. The parameter space was divided into bins with sizes defined by the chosen parameter value range and the number of drawings. This LHC approach aims to ensure a near-random distributed sampling of parameter combinations, so that the whole parameter space is covered and the variability of the data is represented. LHC sampling has been used to calibrate hydrological models (e.g. Singh et al., 2023).

Seven RillGrow input parameters were considered in this way.

- 'DEM base level': vertical distance between the lowest DEM cell elevation and plot outlet level
- 'N for splash efficiency': this determines soil redistribution due to splash i.e. RD-ST (raindrop detachment, splash transport) and RD-FT (raindrop detachment, flow transport) in Kinnell (2001)
- 'Maximum flow speed': a cut-off for cell-to-cell runoff speed which cannot be exceeded
- 'K for detachment': determines soil detachability by flow i.e. FD-FT (flow detachment, flow transport) in Kinnell (2001)
- 'Radius of soil shear stress': this controls the size of the 'patch' over which shear stress is distributed, which controls slumping (gravitational cell-to-cell movement of saturated cells which both exceed a given cell-to-cell gradient and which exceed a shear stress threshold).
- 'Threshold shear stress for slumping'
- 'Angle of rest for slumped soil'

Whilst other input parameters could have been chosen, we focused on these following a first simple Monte Carlo simulation using 3000 runs and testing twelve parameters.

Subsequently, 2000 and 2400 RillGrow simulation runs were made for the laboratory and the field site, respec-
tively, at the high-performance computing centre of the Dresden University of Technology. As it was not practically feasible to save model output at every timestep, selected points in time were chosen, with a higher temporal resolution at the beginning of the simulation and decreasing temporal steps later on. At the field site, in total 36 temporal points (minutes 0.2, 0.5, 1, 2, 3, 5, 7, and afterwards every 3 minutes until 90 minutes) were considered. At the laboratory site 19 temporal points (minutes 0.1, 0.2, 0.5, 1, 2, 3, 4, and then every 2.5 minutes until 30
minutes) were considered.



### 2.5 Objective functions for erosion model evaluation

Erosion model results were compared with measured sediment yield and measured DoDs considering, in total, ten different objective functions and their combinations. The objective functions for different observations were tested with regard to their suitability to calibrate the erosion model. We distinguish between three different spatio-temporal characteristics – i.e. space-time averaged, time-averaged, and area averaged (Table 1) – and different options to calculate comparison metrics – i.e. total change, root mean squared error (RMSE), dynamic time warping (DTW) distance, and normalized Nash-Sutcliffe efficiency (NNSE). Also, observations from different sources, both image-based elevation change models (EC) and sediment yield (SY) measurements, were used (Table 2). Finally, the combination of different objective functions was investigated.

Space-time-averaged data considers total change in measurements, for example change in the total sediment lost during the rainfall simulation experiment, compared to the modelled sediment lost. Space-time averaged EC refers to the cumulative height change measured at the end of the experiments (i.e. the difference as M3C2-PM between the initial and the final time-slice DEM), compared to the modelled EC.

To compare the observed and modelled spatial pattern of erosion, we performed a time-averaged evaluation of the erosion model. This involved two objective functions. The first objective function estimated the pixel-wise difference: i.e. did a direct comparison of the measured 3D model and the simulated soil surface. In order to do this, the observed DEM must be resampled to the same resolution as the simulated raster. Calculated pixel height differences are eventually aggregated to an average value. The second objective function is a dense vector representation (DVR) using a deep learning (DL) method to calculate image embeddings, i.e. an abstract image representation summarized in one vector, to compare them in the latent (i.e. abstract) feature space. This approach is used here to assess the similarity of spatial patterns. The DL approach is less sensitive to offsets in the position of rills (i.e. several pixel differences in the position of the modelled rill compared with the observed rill). The CLIP (Contrastive Language-Image Pre-training: Radford et al., 2021) model was used to transform the images into the feature space because it has shown to be robust across domains: this is especially relevant for our application with height images, which are usually not part of the training datasets. Afterwards, the cosine similarity score is used to derive a value of similarity between the transformed images. To perform this comparison, the DEMs were transformed to an 8-bit 3-channel image, prior to some filtering of strong height outliers to avoid artefacts by keeping only 95% of height values and the remaining 5% (i.e. the largest heights) being replaced by the closest inlier value. The best performing model runs were chosen with regard to the average difference and similarity values of the final model run of each time series, i.e. we did not compare each simulated and observed 3D surface of a series with each other but only the last ones.

The area-averaged data are time series of spatially-averaged (i.e. whole-plot) changes. In case of the EC data, the DoDs based on the M3C2-PM approach were used to estimate the average height change per point in time, thereby always considering the first DEM of the time series as the reference model for the change calculation of the subsequent models. The simulated DEMs were also differentiated using the first model as reference to eventually receive the simulated time series of the modelled EC. To compare the time series, three different metrics were considered. The RMSE is an accuracy metric that calculates the square root of the quadratic mean of the differences between the modelled and measured values. The DTW distance is a measure that tries to find the optimal match





between two sequences and whose remaining mismatch can be considered as an estimate of the time series simi-
larity. It has the advantage that it is invariant to some non-linear behaviour. This metric has been applied to align
complex time series of topographic change (Anders et al., 2021). The third metric is the NNSE, which is the
preferred metric to assess model performance (Batista et al., 2019). It is a measure that relates the modelled error
variance to the measured one. The closer the value to unity, the better the model predicts the erosion. Thus, in total
six metrics were estimated, i.e. the three summarizing time series values calculated for both EC and SY compari-
sons.

The combination of different objective functions was also evaluated. The multi-objective function approach con-
siders eight different combinations (Table 2). For the multi-objective approach, the best models were found by
keeping the models whose metrics are within the top values, e.g. for finding the best ten models the objective
function values were iteratively sorted and evaluated until at least ten models remain within the top values of each
list. We assume that the use of more objective functions enables us to deal better with equifinality (Beven et al.,
2006) due to different processes being captured with different calibration metrics, although only two data sources
were used as input, i.e. images and sediment yield measurements. For instance, the time-averaged spatial similarity
compares the overall appearance of the rill network whereas the RMSE of the time series of elevation change
ideally captures overall erosion within the plot.

**Table 2: Summary of objective functions and their combinations used for the soil erosion model calibration.**

| Single objective function | | | Multi-objective function | |
|---|---|---|---|---|
| Total EC | Space-time averaged | Total elevation change [m] | EC t_series | EC time series assessment with RMSE, DTW and NNSE |
| Total SY | | Total sediment yield [kg] | SY t_series | SY time series assessment with RMSE, DTW and NNSE |
| RMSE EC | Area-averaged | Root mean squared error elevation change [m] | EC & SY t_series | EC & SY time series assessment with RMSE, DTW and NNSE |
| RMSE SY | | Root mean squared error sediment yield [m] | EC t_series & total change | EC t_series and considering total change |
| DTW EC | | Dynamic time warping distance elevation change [m] | SY t_series & total change | SY t_series and considering total change |
| DTW SY | | Dynamic time warping distance sediment yield [kg] | EC & SY t_series & total change | EC & SY t_series and considering their total change |
| NNSE EC | | Normalized Nash-Sutcliffe efficiency elevation change [m] | EC t_series & total change & spatial pattern (no DL) | EC t_series & total change and considering per pixel EC |
| NNSE SY | | Normalized Nash-Sutcliffe efficiency elevation change [kg] | EC t_series & total change & spatial pattern | EC t_series & total change and considering per pixel EC and sim DL EC |
| Per pixel EC | Time-averaged | Per pixel elevation difference [m] | EC & SY t_series & total change & spatial pattern (no DL) | EC & SY t_series & total change and considering per pixel EC |
| Sim DL EC | | Similarity of spatial pattern assessed with deep learning | EC & SY t_series & total change & spatial pattern - all | Considering all single objective functions |

For the field experiment, results from the model runs often showed very strong artefacts. Therefore, a filter ap-
proach was applied that assumed smoother changes of the soil surface, including rills, to automatically detect and
correspondingly exclude these faulty simulation runs. Elevation changes between the first and last DEM, (i.e. the
DoD) of each simulation run were smoothed using a Gaussian filter with a five-pixel kernel. The smoothed model



was then subtracted from the original DoD to identify very strong changes. If the difference was below an arbitrary threshold of 5 cm, the change at that pixel is considered valid. Finally, the ratio between the number of affected and non-affected artefact pixels is calculated and if the ratio is below 0.2%, the simulations are considered not plausible.

**3 Results**

Results are shown separately for the field and laboratory experiments. Animations of elevation changes to the soil surfaces of the field and laboratory rainfall simulation experiments may be viewed using Fig. S1 and Fig. S2 for field and laboratory, respectively.

**3.1 Results from the field experiment**

During 90 minutes of the field rainfall simulation with intensity 140 mm h-1 173.5 kg of sediment was lost from the plot. Total discharge was about 3100 l. Total net height changes measured on the entire plot (including the RoI) were 1.2 cm. At the beginning of the rainfall experiment, a rill began to form at the bottom left of the plot. Rill growth then stopped. Later, a second rill began to form in the bottom centre of the plot and then to cut backwards (i.e. upstream). Growth of this rill slowed after some time. Later, the left rill began to grow again, this time

it continued to cut backwards until the end of the experiment. The formation of wide headcuts was also observed during the experiment, these appear more like terraces and are present across the slope. They retreated upslope slightly but did not evolve into rills.

Large artefacts were developed in many model simulations (Figure 4). These artefacts show very large differences in accumulation and erosion between directly neighbouring pixels, which is not a plausible erosion pattern. How-

ever, some objective functions of the time series still indicated a good fit between modelled and observed data, e.g. when considering the lowest values for DTW EC (7.8 cm) and SY (99 kg).



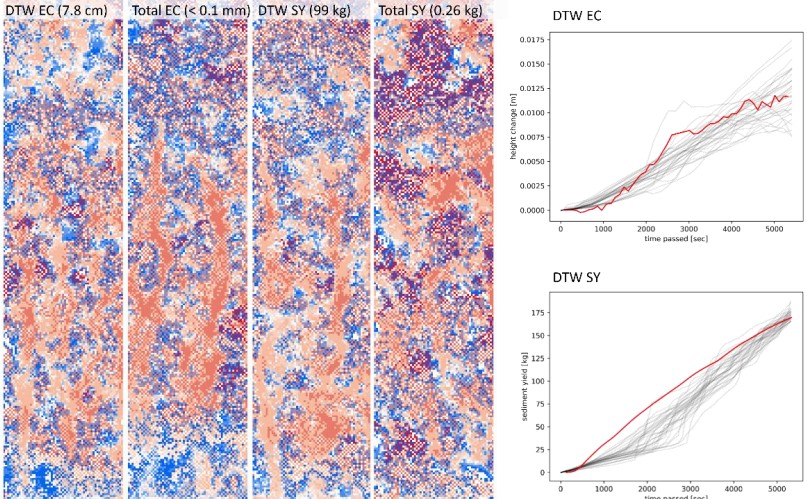

**Figure 4: Although, metrics of objective functions – e.g. time series of DTW of measured (red line) and simulated (grey lines) elevation change (EC) and sediment yield (SY) at field simulation (at right side) – indicate good model fit, large artefacts in final simulated model visible (left side). For legend check Figure 2.**


Also, the plotted time-series of the best 30 simulated model runs, according to the objective function DTW, seemed to indicate a good capture of the averaged EC by the erosion model. An exception was the best model runs found by the DTW SY, in which simulated splash was strongly underestimated (Figure 6). However, total EC (0.1 mm)

and SY (0.26 kg) differences show very good model run fits, again. Still, an inspection of the maps of the final DoDs shows that the best model run, when applying only the EC based metrics (Figure 4 left two plots), i.e. without considering spatial patterns, leads to noisy model runs. This might be due to high positive and negative changes in immediate proximity which are then averaged out during the averaging of the EC. If only SY-based metrics are used, not even rill patterns are visible in the best-simulated best model runs (Figure 4, right two plots).

The good fits in the unfiltered simulation data were therefore found for the wrong reasons, i.e. due to the very strong artefacts, and so they do not reflect finding a good fit of parameters to describe the simulated erosion process.

Using our chosen objective functions alone did not provide sufficient information to automatically assess the plausibility of model output and thus the best input parameters. Only the additional assumption of smooth DEM

changes enabled the removal of these implausible models. When the filter was applied, 279 model runs remained from the original 2000. Different relationships between the objective functions became clear after filtering (scatter plots in Figure 5 and before filtering in Fig. S3:1). In subsequent assessment of the field results, we considered only the filtered data.

The relationship between the different metrics of the objective functions is not linear, except for RMSE and DTW

EC, which indicates that the metrics capture different aspects of soil surface change, including erosion (Figure 5). For example, the model run that has the lowest difference in EC between simulated and observed data does not have to be the same model run with the lowest SY difference. The differences between the EC and SY metrics are more complex, compared to the differences with the same data sources, i.e. camera-based or plot-outlet-measured.



Nevertheless, in general there is a dependence visible between the different metrics. This was favourable, because
we were looking for a model run which ideally fits all objective functions well. The assumption here is the more
diverse the objective functions, the better the different aspects of the model are being captured. Thus, we want to
find the model run with parameter values that fit all constraints defined by the different objective functions. The
objective functions that were chosen for the time series of EC (RMSE, DTW, NNSE for EC) and total change
(EC) fit well, which is also the case for the corresponding SY metrics. The DL based similarity values reveal the
most complex and least obvious relationship to the values of the other functions.

Next, we show the differences between the best model runs chosen according to the different objective functions
which considered total changes (space-time averaged), time-series (area-averaged) and spatial patterns (time-av-
eraged). Figure 6 depicts the final DoD of the best model runs, given the individual objective functions, as well as
their combinations. All variants of the objective functions result in a best model run that appears realistic and that
predicts at least three dominant rills: however, with different lengths, widths and depths. The predicted rills all
reach quite far, compared with observations, into the belt of no erosion towards the upper end of the plot; also,
only two main rills were observed. However, the observations for model validation were only considered for the
RoI, within which the rills did indeed cross the whole length.

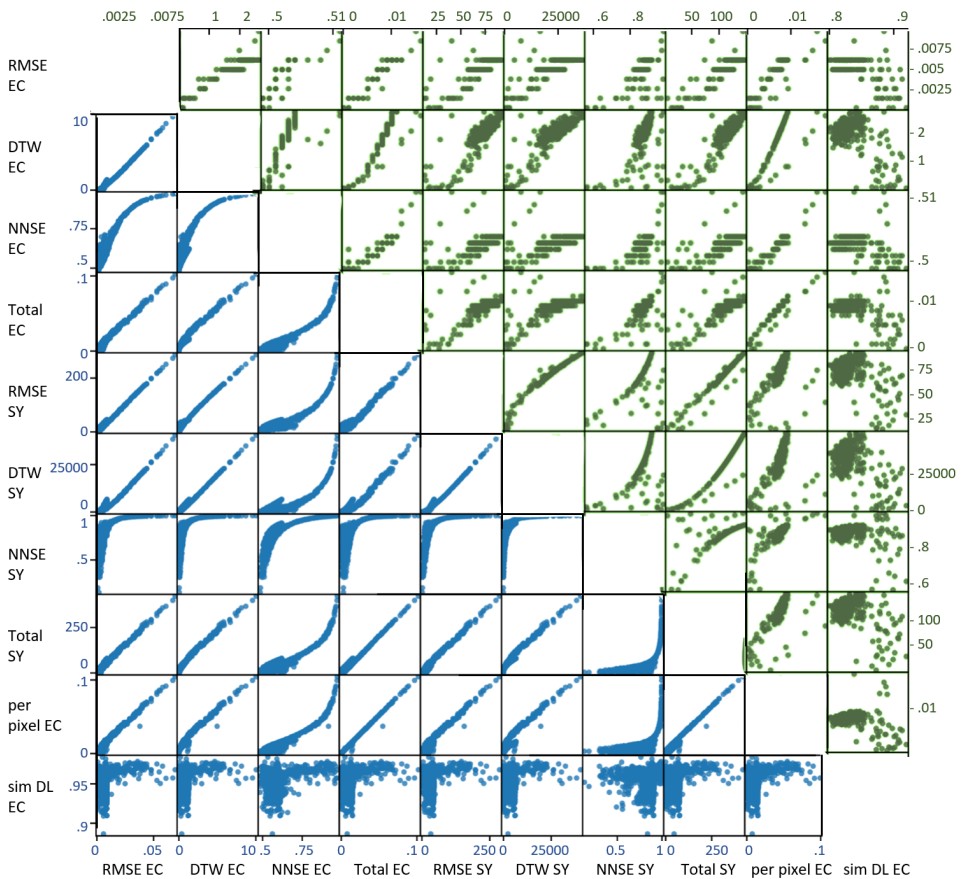



**Figure 5: Scatterplots of values of objective functions at field plot (EC in [m] except for sim DL, which has no unit, and SY in [kg]). Blue scatterplots (lower triangle) correspond to the laboratory experiment and green (upper triangle) to the field experiment.**

The best model runs differ when either single EC- or SY- based objective functions are used for validation. If combinations are considered, there is a difference between using only EC-based metrics and SY-based metrics, or
combinations of EC- and SY-based metrics. The same best model run resulted for DTW EC, total EC and sim DL EC. Thus, EC metrics that consider space-time averaged (total), area-averaged (DTW), or time-averaged (sim DL) characteristics all select the same model run. Further, in that model run no artefacts are present. This is also the case for NNSE SY. The predicted SY for the best EC-metrics-based model run is however about 30 kg (~17%) off from the measured total SY but the predicted and measured total EC is almost identical. In the case of predicted
total EC, the best NNSE SY-based model run is off by 1 mm total EC and by about 24 kg total SY. However, the best model run that was found based on height changes also predicts a more pronounced left rill across almost the whole plot length, which is not the case for the best model run based on the NNSE SY metric. This indicates a potential better performance of EC based metrics to find the best-fitting rill pattern. The best model runs for DTW, total and sim DL EC and NNSE SY do not predict splash or interrill erosion, which does not fit the actual obser-
vation of strong splash effects (Figure 6). More splash is modelled for the metrics DTW and total SY. However, in that model run rill depth is too small. The widest rills are found for the best model run based on RMSE SY and per pixel EC.

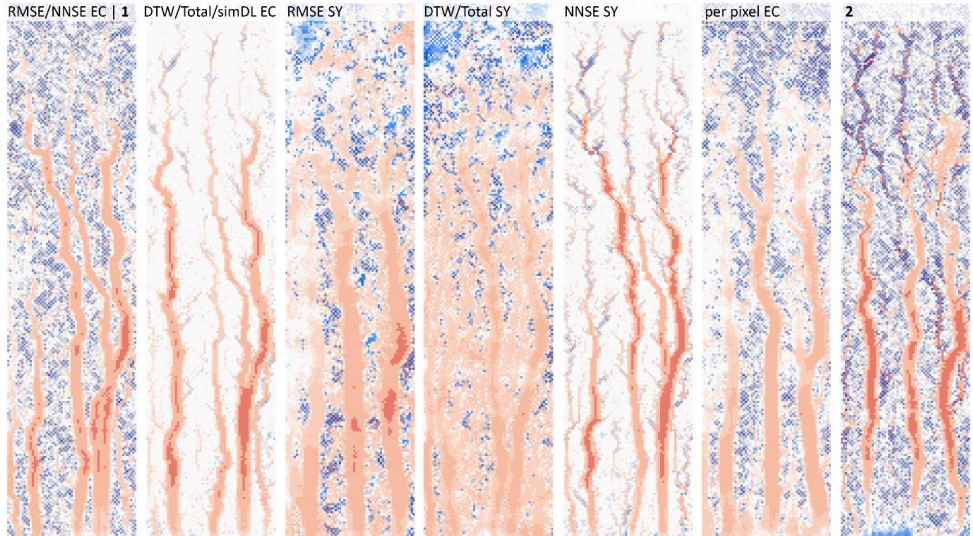

**Figure 6: Simulated height change for single and combination (1: EC t_series, EC t_series & total, EC t_series & total**
**& sp_pattern (no DL), EC t_series & total & sp_pattern; 2: SY t_series, SY t_series & total, EC & SY t_series, EC & SY t_series & total, EC & SY t_series & total & sp_pattern (no DL), all) of objective functions. For legend check Figure 2.**

When considering the combination of objective functions, only two best model runs eventually remained. For all objective functions that consider only EC as parameter (and the various combinations; i.e. EC t_series, EC t_series
& total, EC t_series & total & sp_pattern (no DL), EC t_series & total & sp_pattern) only one model run remained.





The second model run was indicated for all other objective function combinations that consider EC as well as SY (i.e. SY t_series, SY t_series & total, EC & SY t_series, EC & SY t_series & total, EC & SY t_series & total & sp_pattern (no DL), all). Nonetheless it is obvious that these two model runs still contain artefacts, also they do not seem to model splash adequately: this might however be masked by the artefacts. Both model runs predict

three main rills, which are deeper for the second model run, i.e. when EC and SY are considered for model validation.Time series from the erosion model runs (i.e. the area averaged metrics) reveal a good fit by the best 30 model runs at the beginning of the simulation, when changes to the soil surface are still small (Figure 7).

After about 30 minutes the model runs begin to deviate strongly from the observations, independently of the considered objective function (RMSE, DTW or NNSE) for EC. We noted that the erosion model runs were not able

to capture the observed change of erosion rates: after a slow increase of erosion, after about 15 minutes the erosion rate increased steeply then decelerated to a lower rate after about 40 minutes, which then remained nearly constant until the end of the experiment. However, the simulation runs depict more continuous erosion rates. Differences between modelled and simulated time series of SY are not as strong. A better fit in regard of erosion rate and its change during the experiment is visible. Still, the erosion model runs tend to either overestimate or underestimate

erosion throughout the experiment. Very few model runs fitted observations very closely. When considering NNSE no model run fitted, and all the best model runs indicated underestimation.

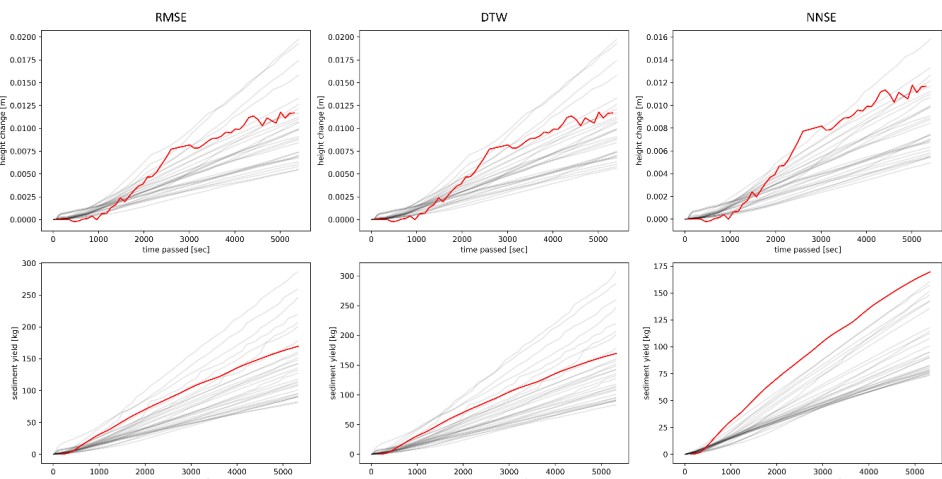

**Figure 7: Time series of elevation change measured with time lapse SfM and of sediment yield measured of plot outlet of best 30 models according to objective functions RMSE, DTW and NNSE at the field plot. Note that the scale of the**
**y-axes differ.**

Considering the observations alone, i.e. EC and SY, the temporal behaviour indicates a difference in both measures. The EC depicts stronger changes of erosion rate during the rainfall simulation compared to the SY rate, which is more stable. This difference might be due to the LoD of the photogrammetry-based data. Thus, only when changes exceed some threshold they are considered, potentially underestimating splash processes in the early rain-

fall phase.

The time-series of the change of the spatial patterns (per pixel EC and sim DL EC) also indicate, at the beginning, a small difference between simulations and observations, since the changes of the soil surface early on in the





experiment are still low. Later on, after about 15 minutes, the patterns become less similar. Figure 8 shows the similarity of spatial pattern for the top 30 model runs according to the averaged per pixel EC and sim DL EC

metrics. The DL-based approach further reveals that the stronger the changes, the closer the similarity values again become: however this is not the case for the per pixel EC metric. This may indicate that it becomes easier to assess the similarity with DL, the stronger the spatial patterns become, due to increasing dominance of erosion rills; whereas during the intermediate phase, elevations of soil surfaces might be ambiguous due to greater noise.

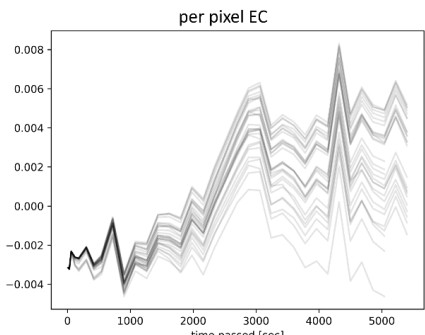 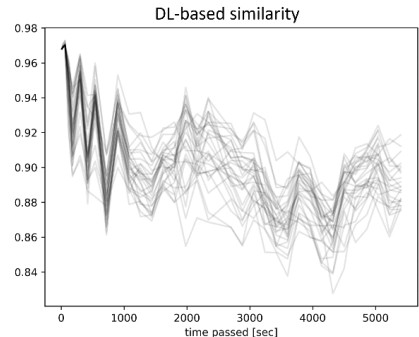

**Figure 8: Time series of spatial pattern metrics for best 30 models according to per pixel EC and DL based EC**
**similarity metric for the field experiment.**

We assessed the best ten model runs considering the different objective function options to evaluate the spread of the model input parameters. Figure 9 shows the parameters within the total parameter range for the best ten model runs according to the multi-objective function approach using all objective functions. It is clear that the parameters do not cluster tightly: equifinality is apparent here, with different combinations of model input parameters giving

similar validation metrics. The spread of parameter values through the whole parameter space remains, mostly for the single objective functions as well as their combinations.



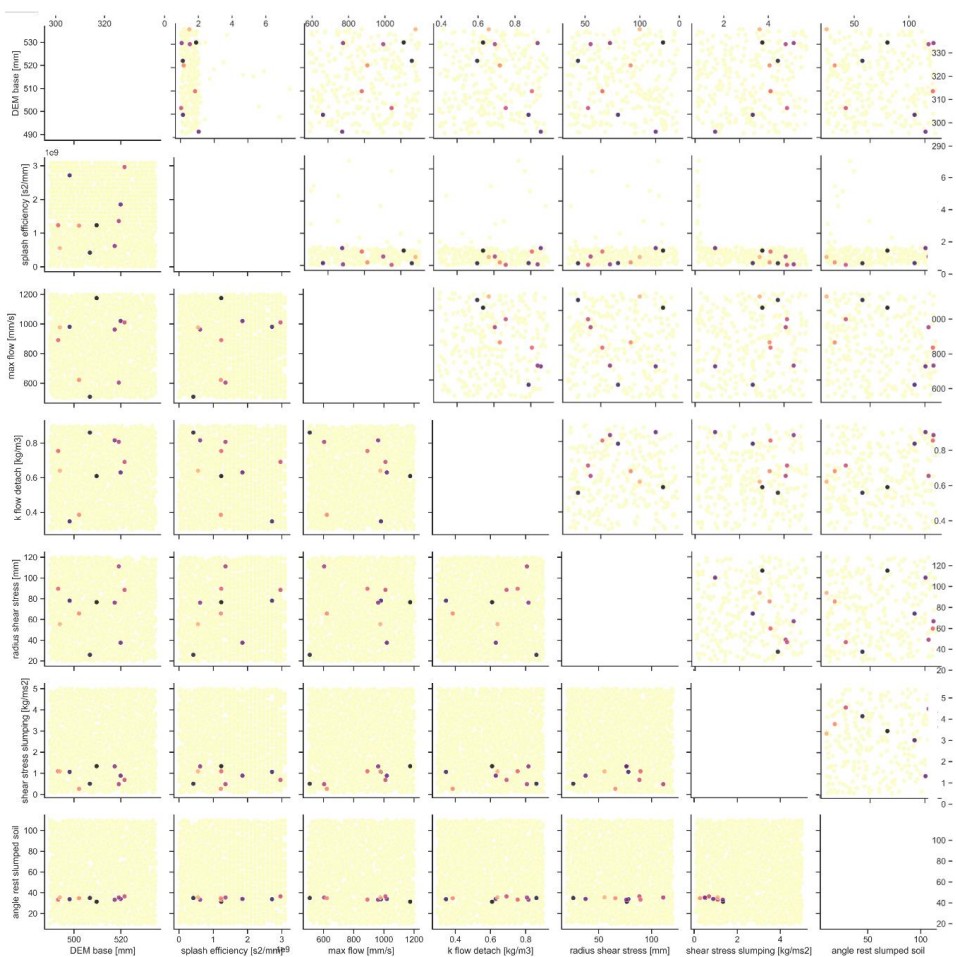

**Figure 9: Parameter values of best 10 models considering all objective functions – lower triangle corresponds to the laboratory experiment and upper triangle to the field experiment.**

However, a closer look at the best ten results for all objective functions indicates decreased ranges of the parameters determining soil detachability by flow and the threshold for shear stress for slumping, because in their combination with the other parameters they are no longer covering the whole of parameter space, but only partial areas, e.g. in the case of "maximum flow velocity" there is an inverse correlation with "soil detachability by flow". In general, very low flow detachment parameters were not considered, which highlights the preference of model

runs with a stronger influence of flow detachment with transport by flow. Filtering of the artefacts from model runs is also visible in the parameter space because the larger splash efficiency values are mostly removed (i.e. in the upper range only very few parameter values remained), indicating that raindrop detachment with transport by raindrop splash is not adequately described by the erosion model since the simulated influence of splash is not confirmed by the observations (i.e. DoDs).



Comparing the parameter ranges for the best 30 model runs, and considering the different options of objective
       function choices, reveals a decrease of the range if more objective functions that implement the EC are used (Figure
       10). Especially, the inclusion of the DL based pattern comparison metric is relevant to narrow down the parameter
       range. However, if the objective functions of EC are combined with SY-based metrics the parameter range
       increases. Also, when considering the SY alone the range remains high. There is actually no difference whether

EC is included or is not. This finding suggests that the calibration of the simulation model can be done using only
       EC observations. This suggestion is reinforced by the circumstance that when evaluating the results for the single
       objective functions, it can already be seen that the best model run for the EC-based metrics has nearly the same
       total SY difference as the best SY-based metrics model runs (i.e. 24 versus 30 kg for the SY and EC-based best
       model runs, respectively).

Assessing the relationship between the model parameters and the metrics of the objective functions (Fig. S3:2-8)
       shows that the choice of lower values for the 'angle of rest for slumped soil' input parameter leads to better model
       performance, whereas larger values have no influence on the performance, i.e. no large changes in regard to the
       metrics can be seen, either because they remain high for the time series metrics or because they remain at a nearly
       constant value for the area and space-area averaged metrics. With regard to the parameter choices for 'DEM base

level', 'radius of shear stress', and 'splash efficiency', no influence on model performance is visible when looking
       at the different metrics. Further, it appears that the larger the choice for the values of the model input parameters
       for 'flow detachment' and 'maximum flow velocity', a slightly better performance of the model runs is given when
       considering the SY-based metrics. However, this relationship is not as clear for the EC-based metrics. Higher
       'shear stress slumping threshold' values lead to better erosion model performances, when considering EC metrics,

but the performance worsens if SY metrics are considered.

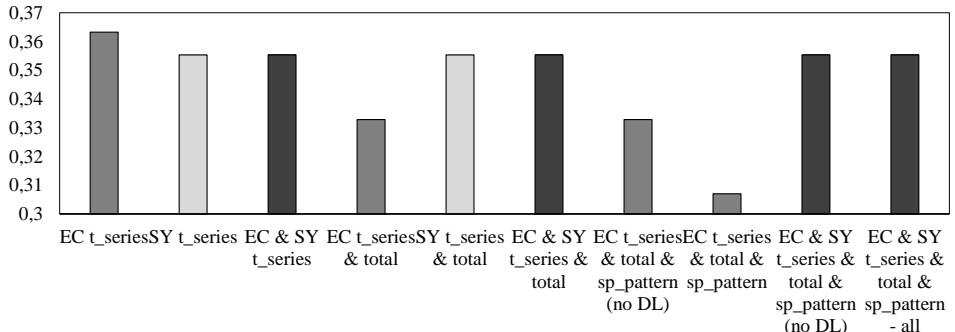

**Figure 10: Range of parameters of best 30 models for different combinations objective functions at the field
experiment. Note that y-axis starts at 0.3. The column colours correspond to the consideration of EC, SY or SY and
EC.**

None of the model runs predicted the observed rill pattern well, i.e. neither the number of rills (two) nor their
       locations were adequately modelled. However, some of the model runs did predict the rill at the left plot side and
       the rill in the middle of the plot. Nevertheless, no model run indicated that the rill at left would become more
       dominant later during the rainfall simulation and that the rill in the middle would be dominant in the beginning of
       experiment and then would stop growing.



## 3.2 Results from the laboratory experiment

During the 30 min rainfall simulation in the laboratory a total 80.8 kg of sediment was lost from the plot, and the corresponding discharge was about 395 l. The total net height change measured on the plot was 1.6 cm. Thus, although the rainfall simulation intensity was the same as the field experiment, a larger negative height change (about 25%) was seen, as well as greater sediment yield (considering that on the laboratory plot, nearly half the SY had already been lost after only one third of the total experiment duration of the field plot). Discharge amounted only to about 8% of the field runoff. These differences are mainly due to the significantly higher slope gradient for the laboratory experiment. The animation (Fig. S1) of elevation changes during the field experiment shows the formation of a rill network which is markedly different from the nework developed during the field experiments. At first, a dominant rill forms at the lower right plot side. A second main rill then develops on the lower third of the plot, at the left side. After about 15 minutes, the left rill merges with a third rill that began to grow at the bottom of the plot after about ten minutes, cutting upslope until it meets the other rill. At the end of the experiment an intricate dendritic rill pattern is observable, especially in the upper third of the plot which is also dominated by sheet erosion, draining into the two main rills. The rill network formed in the first 15 to 20 minutes and then changed little, mainly just deepening.

Again in contrast to the field experiment, in the laboratory rainfall simulation no strong artefacts were observed in the erosion model output. Therefore no filtering was necessary and all 2400 model runs could be used for the evaluation using the different single objective functions (Figure 11). Although the best model runs of the various calibration functions predict the observed filigree rill pattern and/or a few single strong rills, none of them model the strong sheet erosion. In the best model runs based on the area-averaged parameters (RMSE, DTW, NNSE) four dominant rills are predicted that are wide and deep, and situated across the whole plot region. Considering total SY for calibration, yields three obvious rills. The right rill fits the observation data well as it is also bifurcated. However, none of the SY-based best model runs capture the overall fine rill pattern. The best model runs for the space-time- and area-averaged EC-based metrics also predict wide rills covering the lower plot region, which is closer to the camera-based observations. The observed total SY is underestimated by 12 kg (DTW, RMSE, total EC) to 16 kg (NNSE EC), whereas the residual of predicted total SY for the best SY-based model runs range between 1 kg (DTW SY) and 3 kg (NNSE, RMSE SY). The best model run for the objective function considering per pixel EC is the only one that predicts two dominant rills. However, these are still too wide and short compared with observations. The best model run for the second metric focusing on spatial pattern, i.e. sim DL EC, predicts the formation of many small rills. Thus, it is closest to the observed dendritic rill pattern, but the rills are too shallow and this model run completely misses the observed sheet erosion.



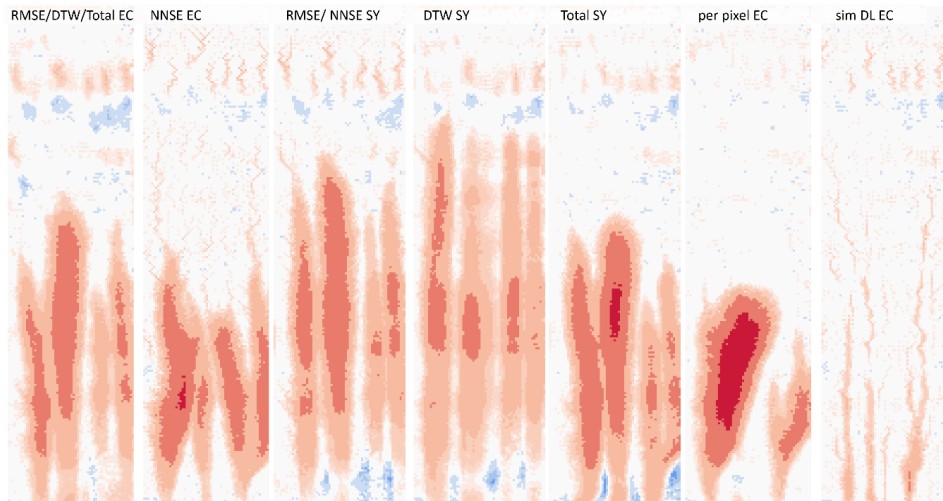

**Figure 11: Simulated height change of the best model in regard of the single objective functions. For legend check Figure 2.**

When combining the objective functions, it is not possible to find a single best model run that fits the observations
well (Figure 12). In case of EC and SY time series based parameters, model runs predict patterns with rills that are
too wide and too long, and which cover the whole plot. The combination of objective functions considering only
EC-based metrics, including the spatial pattern, indicate model runs with rill erosion dominating in the lower to
the middle part of the plot: however these are still too long, and there is a belt of no erosion at the upper plot region.
If SY-based metrics are combined with EC-based ones, then predicted rills are shorter, but still do not resemble
the observations as the rills remain too wide. Overall, in the camera-based observations in the upper plot region
strong sheet erosion is visible, which is not predicted by the erosion model.

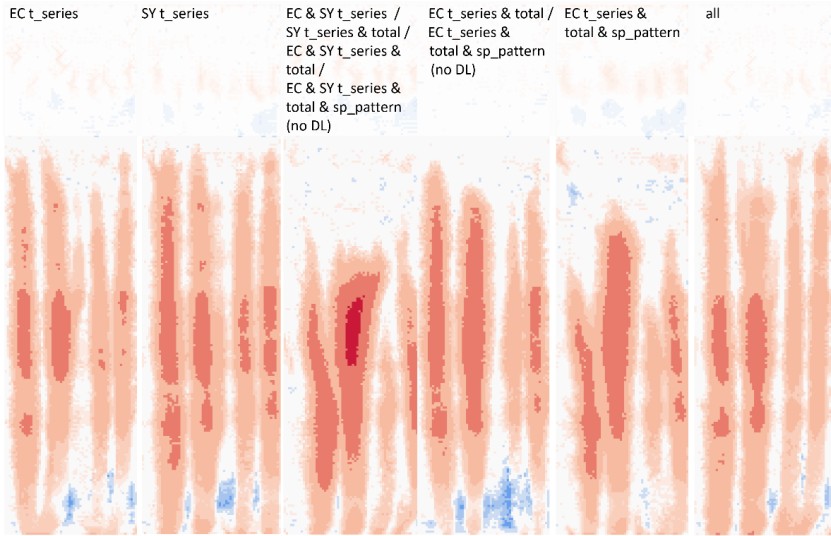





**Figure 12: Simulated height change of the best model in regard of the multiple objective functions. For legend check Figure 2.**


Relationships between the different calibration metrics are more obvious for the laboratory experiment (Figure 5). Correlations between all objective function values, except for NNSE and the DL based spatial pattern comparison, are clearly linear. The relationship between the values of the NNSE SY and EC and the remaining metrics is obvious and non-linear. The DL based similarity metric alone reveals an unclear relationship to the values of the other functions.


When assessing the temporal behaviour of the best 30 model runs according to the area-averaged metrics, the EC-based approaches reveal model runs that fit well or tend to underestimate soil erosion in the middle period of the experiment (Figure 13). The best model runs according to the RMSE and NNSE EC do a better job of predicting height change over time, while the DTW EC based models give a best fit at the beginning and end of the experiment


(as it is expected due to the way in which the DTW is calculated). The NNSE EC based simulations show the largest variance between the best 30 model runs at the end of the experiment. In contrast to the field experiment, the rate of change for the observed EC and SY is similar in the laboratory rainfall simulation, which may result from more intense change at the beginning of the experiment, leading to crossing of the LoD threshold early on. The best model runs according to the SY-based calibration values also fit the observations well. The RMSE and


NNSE SY-based best model runs overestimate erosion, especially at the beginning, but also slightly throughout the rainfall experiment. The NNSE-based best model runs scatter strongly towards the end of the experiment. The best model runs according to the DTW SY metric reveal a stronger variation in regard of the temporal behaviour of the sediment yield with a few model runs strongly underestimating erosion.

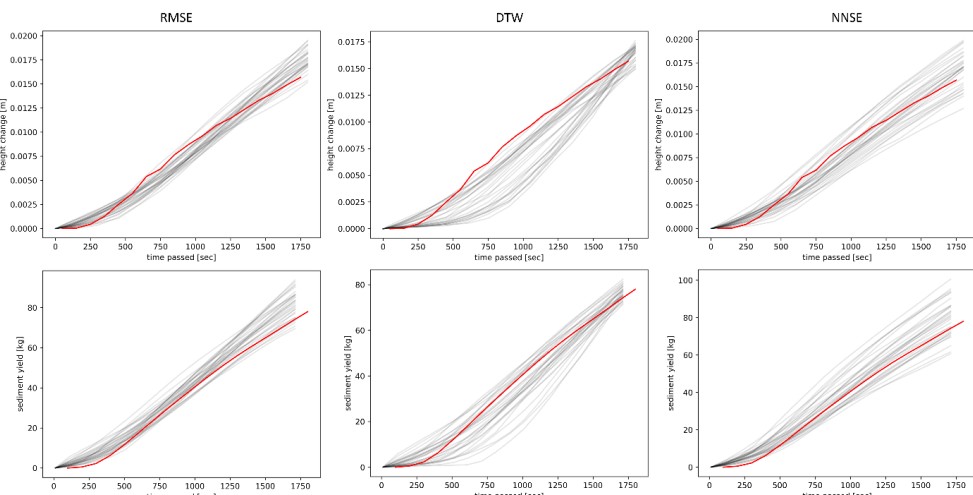


**Figure 13: Time series of elevation change measured with time lapse SfM and of sediment yield measured of plot outlet of best 30 models according to objective functions RMSE, DTW and NNSE at the laboratory plot. Note that the scale of the y-axes differ.**

In both field and laboratory experiments, the spatial pattern metrics indicate a high agreement between model results and reality at the beginning of the experiment. Later in the experiments these begin to diverge, when


considering the per pixel EC (Figure 14). However, the best model runs according to the DL-based metric already




show an increase in the deviation between observation and model very early in the experiment; towards the end the differences decrease again. This is not surprising because the best 30 model runs were chosen using the difference of last model runs in the time series. In general, model runs of the laboratory experiment were more similar to each other than was the case for model runs of the field experiment.

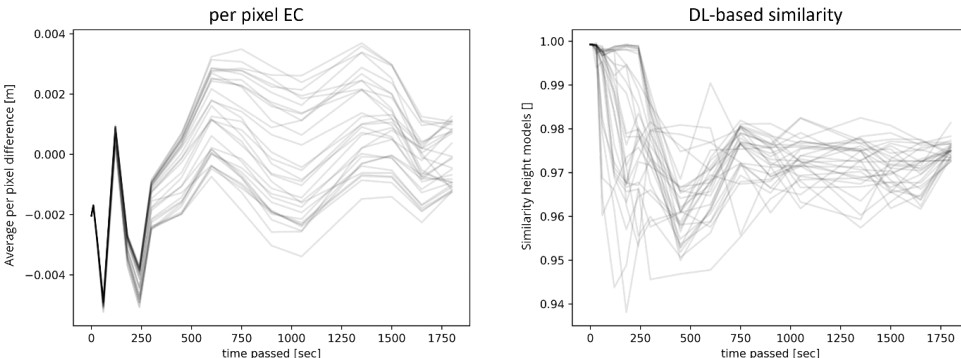


**Figure 14: Time series of spatial pattern metrics for best 30 models according to per pixel EC and DL based EC similarity metric for the laboratory experiment.**

Compared to the field experiment, the combination of objective functions enables a more obvious narrowing down of the range of parameter values with regard to the 'threshold of shear stress for slumping' and the 'angle of rest

for slumped soil' input parameters, both in favour of lower values (Figure 9). For the input parameter 'distance to the DEM base level', lower values are chosen for the best ten model runs. The remaining parameters are spread across the entire range.

The best 30 model runs had a lower range of parameters when all EC based metrics and when all objective functions were considered (Figure 15). The remaining combinations of objective functions reveal higher but

similar ranges and no difference can be seen if EC and/SY is considered.

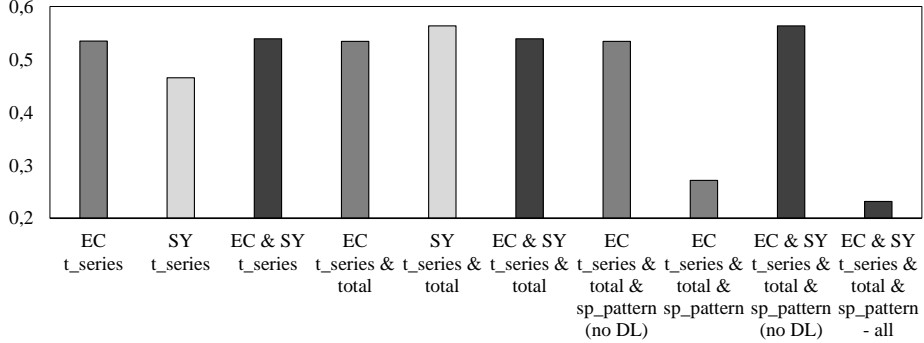

**Figure 15: Range of parameters of best 30 models for different combinations objective functions at the laboratory experiment. Note that the y-axis values start at 0.2.**

For the laboratory experiment, relationships between the erosion model input parameters and the metrics of the

objective functions indicate similar characteristics to the field experiment, except for the SY and considering the 'threshold of shear stress for slumping' parameter (Fig. S3:9-15). Overall, no model run did a good job of





predicting the response of the soil in the laboratory rainfall simulation experiment to the multiple and interacting processes of soil erosion. Rills were too long and wide and no finely-detailed rill pattern was predicted.

## 4 Discussion

This evaluation of a single soil erosion model used three approaches to spatio-temporal averaging (Table 1). While other erosion model evaluations (e.g. Favis-Mortlock et al., 1996; Jetten et al., 1999) have considered multiple erosion models, few if any previous erosion model evaluations have considered all three spatio-temporal averaging approaches. The most commonly-used approach involves space-time averaging: a simple comparison of measured and modelled runoff and sediment yield at the end of the plot (or at the catchment outlet) and at the end of the

period of observation. Sometimes this is supplemented by comparisons of measured and modelled time series, constructed from additional plot-end (or catchment outlet) measurements during the period of observation. It has been, and still is, much less common to compare measured and modelled spatial patterns of erosion.

From the perspective of the historical development of erosion models (e.g. Nicks, 1998) it is unsurprising that model evaluations have concentrated on simple spatio-temporal averaging approaches. The USLE (Wischmeier,

1976) and subsequent models based upon the USLE (e.g. MUSLE: Williams, 1975; RUSLE: Renard et al., 1991; USLE-M: Kinnell and Risse, 1978) are only capable of generating results which are averaged over both time and space. Subsequent, more process-focused, erosion models such as WEPP (Nearing et al., 1989) or Erosion-2D (Schmidt, 1991) introduced a 2D spatial element: a hillslope profile. Later came models with an explicit spatial focus, such as LISEM (de Roo et al., 1996) and Erosion-3D (Schmidt et al., 1996). The evaluation of erosion

models has therefore lagged somewhat behind the development of the models themselves.

An important finding from this study is that is that each averaging approach, as exemplified by each group of objective functions, illuminates different aspects of erosion model performance, as discussed below. No single objective function is capable of identifying all strengths and weaknesses of the model tested. Thus, as we appreciate more strongly the interacting temporal and spatial complexities of soil erosion – whether at the process-dominated

plot scale, as in this study, or the connectivity-controlled catchment scale (Favis-Mortlock et al., 2022) – and incorporate representations of this complexity in future erosion models, it is clear that we will need approaches to model evaluation of the kind described in this study.

A particular challenge, when comparing measured and modelled patterns of rill erosion, results from the problem of small spatial offsets in rill location. A modelled rill network might, as judged by eye, be very similar to an

observed rill network. But a simple pixel-by-pixel comparison of measured and modelled DEMs could still give poor results since rills may well be spatially offset between the two DEMs, perhaps by very small distances. Additionally, measured and modelled rill depths may differ. There is much less of a problem when considering interrill changes, since these are dispersed across the whole of the eroding area and thus averaged values can be used. AI-based similarity objective functions (Radford et al., 2021), as used in this study, could provide a potential

solution to this issue. Such objective functions can give a clear measure of the fit between model and observation even if the rills are not at identical locations.

We found the best field experiment model runs using either EC-based single objective functions, i.e. DTW, total and sim DL, which suggests that in some cases an ensemble of objective functions might not be needed. However,





it also became clear that there is a sensitivity towards the choice of the single objectives, because for the metrics
RMSE, NNSE and per pixel EC, another best model run was found. Finding the best model runs using only EC-based measures suggests that the erosion model calibration might be possible without using sediment yield observations. This has important implications for the usage of our approach to calibrate models also at larger scales, i.e. when time series of catchment data via UAV or aerial measurements may not be available.

The calibration metric DTW is not the most suitable for the SY-based measurements, for the field experiment.
However, if it is EC-based, the best model run was found to outperform the fit of the RMSE and NNSE EC-based metrics. This may be due to the averaging of the SY changes with no spatial consideration, whereas the averaged EC-values are still based on spatial measurements. The best temporal behaviour of soil surface change (EC) or SY were captured before the artefact filtering, i.e. the DTW, RMSE and NNSE values and corresponding plotted time series indicated a very good model fit. It is clear that splash redistribution is indeed an important process in our
two experiments. But the erosion model was not able to represent this process adequately. Further work is needed to avoid creating the artefacts that make the model outcome implausible.

When using time-lapse SfM-photogrammetry to measure soil erosion it must be considered that erosion-masking processes, such as soil compaction and settling, can lead to faulty erosion measurements (Kaiser et al., 2018, Epple et al., subm.). In these two experiments, such processes are assumed to be negligible due to the application of very
strong rainfall events on relatively compacted soils (soil bulk density of 1.23 tm-3). Similar considerations apply when using laser scanners, e.g. Wang and Lai (2018).

From the perspective of future work: this study has clearly indicated weaknesses in some process representations within RillGrow, particularly splash redistribution. Work is ongoing to improve this. In addition, the computational needs of the model meant that the multiple model runs required by the current study used a great deal of computing
power and time. A parallel-processing version of RillGrow is being developed.

In addition to considering changes in DEM elevation, it might also be useful in future studies to consider measurements of ponding and runoff forming at the soil surface (Zamboni et al., 2024) or spatially distributed velocity quantities (Wolff et al., 2024). These can be used to provide further calibration/evaluation opportunities focusing on the hydrological, rather the sedimentological, processes.

Another approach for future work could focus on the assessment of weighting the different objective functions. This is of interest because this study reveals that some objective functions are more important than others, such as when considering spatial patterns versus time series of averaged change metrics (e.g. DTW EC versus sim DL EC and their combinations, i.e. EC t_series & total & sp_pattern at the field experiment). We tested a weighted error after standardizing the objective function values. However, such a uniform approach did not produce good results
for the field experiment, i.e. the best model runs contained only very small rills. Thus the optimum weighting of the different functions is not known. Another aspect for improvement is the consideration of the parameter distribution. The best ten or thirty model runs revealed that the output is not one set of parameters, but actually a set of parameter distributions. However, these distributions are not independent, i.e. if one parameter is chosen, it means that a specific other needs to be drawn. Therefore, in the future, conditional drawings should be considered
(i.e. Bayes principle).



## 5 Conclusions

This study advances the calibration and evaluation of soil erosion models by considering various objective functions that consider spatio-temporal aspects differently. Several thousand runs of the erosion model RillGrow were performed with parameters drawn approximately randomly, by use of a Latin Hyper Cube approach. Outputs from these model runs were compared to sediment yield measured during field- and laboratory-based rainfall simulation experiments, and model outputs were compared to SfM photogrammetry-derived observations, i.e. representations of soil surface change with spatial resolutions of few cm and temporal resolutions of 10 seconds. Ten calibration metrices were used to find the best-performing model runs.

Results highlight the need for more sophisticated evaluation techniques that go beyond traditional space-time averaging methods. Different spatio-temporal averaging approaches illuminate different aspects of model performance, indicating that no single objective function could fully capture the complexities of erosion processes. The study also identified challenges in model validation, such as the issue of spatial offsets in rill locations, and suggests AI-based similarity functions as a potential solution. Additionally, the study clearly identified limitations in the process representations within the version of RillGrow used in the study, particularly regarding splash erosion. Such a finding is very useful with regard to prioritising ongoing refinement of the erosion model.

The exploration of alternative calibration metrics and the potential for parallel-processing to address computational demands illustrates the evolving landscape of erosion modeling. Our findings suggest that future research should focus on refining objective functions that also consider novel observations of the soil erosion processes, as such observations are likely under future global change. Future research should also consider parameter distributions to improve calibration and validation outcomes.

Overall, these results strongly emphasize the need for more nuanced evaluation of erosion models, including the incorporation of spatial pattern comparison techniques. This is necessary to provide deeper understanding of any erosion model's capabilities. Only with such improved model evaluations will we be able to adequately develop and evaluate a future generation of soil erosion models, which will be vital tools in forecasting and managing the impacts of future global change

### List of acronyms

| | |
|---|---|
| CA | Cellular Automaton |
| CLIP | Contrastive Language-Image Pre-training |
| DEM | Digital Elevation Model |
| DL | Deep Learning |
| DoD | DEM of Differences |
| DTW | Dynamic Time Warping |
| DVR | Dense Vector Representation |
| EC | Elevation Change |
| FD-FT | Flow Detachment, Flow Transport" (Kinnell, 2001) |



| | |
|---|---|
| GCP | Ground Control Point |
| LHC | Latin HyperCube |
| LoD | Level of Detection |
| NNSE | Normalized Nash-Sutcliffe Efficiency |
| PM | Precision Map |
| RD-FT | Raindrop Detachment, Flow Transport (Kinnell, 2001) |
| RD-ST | Raindrop Detachment, Splash Transport (Kinnell, 2001) |
| RMSE | Root Mean Square Error |
| RoI | Region of Interest |
| SfM | Structure from Motion |
| SLR | Single Lens Reflex |
| SY | Sediment Yield |
| UAV | Unmanned Aerial Vehicle |

**Data & Code availability:** The raw data and source code for model evaluation are available here: https://doi.org/10.25532/OPARA-602. The source code of RillGrow can be accessed here: https://github.com/davefavismortlock/RillGrow.

**Author contribution:** All authors contributed greatly to the work. AE: conceptualization, methodology, investigation, writing (original draft), figures, data acquisition & processing, funding acquisition; DFM: conceptualization (support), investigation (support), writing (original draft); OG: data acquisition & processing SfM data; MN, TL, PK: data acquisition & processing soil data, rainfall simulator setup, writing (review and editing). All authors have read and agreed to the published version of the paper.

**Competing interests**: The corresponding author has declared that none of the authors has any competing interests.

**Acknowledgements**

We would like to thank the HPC of the Dresden University of Technology (ZIH) for their computer resources. Furthermore, we are thankful for comments and discussions by Anne Bienert, Lea Epple and Jonas Lenz and for field support by Pedro Zamboni. The project was founded by the German Research Foundation (DFG): project number 405774238 and by the Ministry of Agriculture of the Czech Republic by grant number QK22010261.

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
