# Peer review of "Using 3D observations with high spatio-temporal resolution to calibrate and evaluate a process-focused cellular automaton model of soil erosion by water"

_EGUsphere, 2024_

## Author Comment (AC1)

This manuscript is precisely what we need in soil erosion modelling: innovative, nuanced approaches to improve model evaluation and an honest assessment of model performance. In the specific comments below, I made several questions that came up while reading the manuscript and some suggestions to hopefully improve the paper. Specifically, I recommend adjusting some of the modelling terminology (e.g. the usage of terms such as calibration, validation, and evaluation) and reducing the focus on identifying a 'best model run'. Moreover, the figures generally could use some improvements. I expand on these topics in specific comments below.

Thank you very much for your most encouraging statement and thank you very much for your detailed comments and suggestions, which helps a lot to improve our manuscript. We are addressing each of your comments below.

Specific comments
Abstract: Here and in the introduction you could highlight the novelty of your work. To my knowledge, this is the first time such high spatiotemporal resolution data is used for evaluating erosion models.

We add one more sentence to state this more specifically and in general revise the abstract.

L50-75: I found the model description a bit on the long side for the introduction. I would consider moving most of this to section 2.3 in the methods.

We remove some part to the methods, i.e., lines 62-75.

L74: Please define the abbreviations RD-ST and RD-FT.

Done.

L105: "Validate" and "evaluate" seem to be used interchangeably, but these terms can signify different meanings (Oreskes, 1998; Oreskes et al., 1994). Beven and Young (2013) suggest avoiding the term "validation" in hydrological modelling.

We agree and carefully revise the manuscript to more precisely consider the suitable term.

L121: Consider rephrasing to: "Ten objective functions were considered to *calibrate* model parameters".

Thank you for the suggestion. Done.

L122: By model runs, do you mean model realisations, i.e. "one random sample taken from the set of all possible random samples in a Monte Carlo simulation" (Beven, 2009)?

Yes, indeed we mean model realisations. Each run corresponds to a randomly drawn set of model parameters. These sets were beforehand drawn considering the LHC sampling to enable the coverage of the whole range of the parameter space. After the parameter samples were collected, we ran the simulations and picking each time another of the sets, i.e. 2000 and 2400 sets for the field and laboratory run, respectively.

I missed a stronger statement about the importance and novelty of your work. What you have done is innovative and exciting and creates new possibilities for testing erosion models.

Thank you to support our work. We add another sentence to highlight the novelty of our work.

L167: How did you choose the DEM resolution?

The resolutions were chosen considering the runtime of the erosion model, i.e. getting the results in a reasonable time for so many runs even though a high performance computer had been used, and still having the resolution as high as possible to capture different processes.

L168: Why do the time-lapse data have finer resolutions than the DEMs used as input for the model?

The resolution of the time-lapse data is actually the maximal resolution we could achieve in our experiments. Thus, as much as possible information is contained. However, the model runs in RillGrow needed a down-sampling to ensure results in a feasible amount of time. Thus, for direct comparison of model run outputs and measured models with the cameras, we needed to ensure that both models have the same resolution, using a down-sampling approach for the higher resolved camera data. But when we compared averaged values, e.g. the average elevation change per time step, we considered the original resolution of both models because we want the erosion model to be calibrated such that they predict the actual surface changes as close as possible to reality approximated with the camera data the better the higher the resolution.

L170: What is M3C2-PM?

This is the multi-scale cloud-to-cloud comparison approach considering precision maps. How the algorithm works is explained in a bit more detail in the following sentences of the same paragraph. We would prefer to refer to the original paper for more detail to avoid increasing the length of an already pretty long manuscript.

L170-180: It's great to have this spatially distributed error estimate for the point clouds. Have you considered using this as part of the model evaluation process, i.e. for defining limits of acceptability of model error (Beven, 2018)?

No, we have not considered this, yet. However, this is a very nice idea and a great next step to further improve the model evaluations considering time-lapse 3D data of change.

I appreciate the narrative model description, but having a list of model equations in the supplement would be very helpful. Please add this information.

This is added to the Supplementary Material.

L204: Please check if this formulation is correct: "If a wet cell's sediment load is less than the transport capacity, then soil is eroded from the cell using a probabilistic detachment equation by Nearing (1991)".

Shouldn't you first calculate soil detachment for a given cell, sum it to the sediment load delivered to this cell by upstream cells, and then compare it to the transport capacity of the overland flow for this given cell to estimate the amount of sediment routed downstream? That is, why would soil detachment (and not transport) be dependent on the transport capacity of the overland flow? Maybe I misunderstood something – please clarify.

RillGrow assumes that overland flow always carries as much sediment as the flow's transport capacity will support. If there is less sediment being transported, then sufficient soil is eroded to bring the quantity of transported sediment up to the maximum that can be transported. This seems intuitively plausible: if (for example) flow speeds up as it moves over a steep (compared with upstream) cell-cell slope, then it will gain an increased ability to erode.

Here is a revised, and hopefully clearer, description: „Following flow routing, each wet cell has a sediment load which has been received from the adjacent upstream cell (or cells). The model then calculates the transport capacity for the cell using equation 5 from Nearing et al. (1997). If the sediment load exceeds the transport capacity, then excess sediment is deposited assuming a linear function of the difference between sediment load and transport capacity (equation 12 in Lei et al., 1998). If the sediment load is less than the transport capacity, then soil is eroded from the cell using a probabilistic detachment equation by Nearing (1991): this represents FD-FT in the Kinnell (2001) classification of erosion subprocesses."

Assuming that it is clearer, the above is substituted in the paper's text.

L231: I found it strange to calibrate the parameter 'DEM base level', as this is a measurable quantity that would not need to be estimated via calibration. Can you explain your rationale here?
RillGrow's early runs used data from laboratory flumes. As is well known, dealing with the soil on the lowest edge of the flume (i.e. the edge where flow leaves the flume) can be problematic. If the soil in the flume is too deep, whole chunks of soil can detach and fall from this edge, If the soil in the flume is too shallow, the exit rill cuts down until it hits the bottom of the flume and can of course go no further: this artificial baselevel may then constrain erosion on the whole of the flume. Something similar occurs with the model. If soil is too deep (i.e. the baselevel is set very low) then the exit rill can (and usually does) very quickly cut down, unrealistically deeply. If the soil is too shallow (i.e. baselevel is set too high) then the bottom of the exit rill hits this baselevel, and this then constrains erosion on the whole of the flume. The ideal depth of erodible soil (i.e. distance down to baselevel) in both laboratory reality and model simulation is somewhat the result of trial and error.
Incidentally, it is an interesting kind of "validation" to find that both reality and model have similar problems in this regard. A kind of equifinality, we assume.

L242-243: Based on this initial simulation, how did you choose the parameters for calibration? Based on some kind of sensitivity measure?
Indeed, we ran the first models with 12 parameters instead of ten using a simple MC simulation with 3000 draws of parameters. Thereby, however not yet considering LHC sampling. This was done to get an initial idea for the rough borders of parameters to be drawn from and to evaluate if all parameters are needed. The initial range of parameters was chosen based on the experience of the co-authors, working especially in the field of erosion modelling and on the experience of the model developer of RillGrow.

What was the parameter space sampled with the Latin hypercube simulation? Please give ranges (assuming you were sampling from uniform distributions) for each calibrated parameter.
The ranges were chosen the following (we add the information to the supplementary):
Field experiment
Constant n for splash efficiency (sec**2/kg.m): 5e10 - 2.5e13
Base level e.g. distance below lowest DEM point to flume lip [cm]: 5 - 50
Maximum flow speed (mm/sec): 500 - 1100
Constant k for detachment (kg/m**3): 0.3 - 0.9
Radius of soil shear stress 'patch' (mm): 20 - 120
When saturated, threshold shear stress for slumping (kg/m s**2): 0.1 - 5
Angle of rest for saturated slumped sediment (%): 10 - 110
Laboratory experiment
Constant n for splash efficiency (sec**2/kg.m): 5e10 - 2.5e13
Base level e.g. distance below lowest DEM point to flume lip [cm]: 5 - 50
Maximum flow speed (mm/sec): 500 - 1200
Constant k for detachment (kg/m**3): 0.3 - 0.9
Radius of soil shear stress 'patch' (mm): 20 - 120
When saturated, threshold shear stress for slumping (kg/m s**2): 0.1 - 5
Angle of rest for saturated slumped sediment (%): 10 - 110

L254: How did you assess the suitability of a function for calibrating the model?

The testing is meant in the regard that we wanted to evaluate whether the calibration function (e.g., RMSE EC) is able to find the set of model parameters that best predict the rill erosion, i.e., when I optimize my model with such a function, do I find a minimum/maximum, e.g. the lower the RMSE EC the better the model parameters for the prediction. And this assessment was done by considering the final DoD and comparing it visually with the measured change and doing the same for the sediment yield as well as comparing the values of each of these calibration functions to each other to assess whether some of them show a very high correlation.

L269-280: How is the DL metric interpreted? The higher the value, the greater the similarity?
Yes, indeed, the higher the value the higher the similarity (ranging from 0 to 1) as the similarity corresponds to the cosine of the angle between the two latent feature vectors that ideally encoded the pattern of each of the DoDs.

L310: Why did you smooth out the DoDs and not the simulated DEMs?
The idea was to remove only pixels that revealed very strong changes of change within short distances, i.e. this could be considered as considering roughness of the DoD. And as we assume more smooth and consistent change in a local neighborhood, we smoothed the DoD and then subtracted the original one. This is actually an adoption from an idea in Onnen et al. (2020). We could have used the DEM as well as the surface was quite smooth, but for more rough surfaces, e.g. the soil was freshly ploughed, the DoD would be more suitable.

Explaining the methods employed in section 2.5 was difficult, but I think you did a good job. Still, I have some questions/comments:
- Is it possible to compare the measured sediment yield from the plot to the soil loss calculated from the DoDs (e.g. Cândido et al., 2020)?
- Yes, that would be possible if we consider the soil bulk density (and do not assuming erosion masking processes such as settling and swelling/shrinkage; see Kaiser et al., 2018). However, since RillGrow directly provided the modelled sediment yield and elevation change, we can use the information directly for the assessments.

- In Table 2, the column named "single objective function" for the space-time averaged and time-averaged model evaluations just mentions the type of data used for model testing, e.g. Total EC and Total SY. But I reckon you calculated e.g. the RMSE and the NSE for the total EC and total SY data. Is this correct?
- No, we did not calculate the RMSE and NSE for total EC and SY because here we refer to the overall height change and sediment yield resulting in just on value. However, we note why this might be confusing as it should be rather referred to as Total difference of SY and EC. We clarify this in the table in the revised manuscript. The RMSE, NSE and so on was calculated for the scenarios when we had several difference values to the reference.

- In lines 286-292, you explain the metrics (the objective functions?) used for model evaluation for the time-averaged data. But they are also used for the other data, right? This got a little confusing.
- Indeed, the metrics are the output of our objective functions to assess the model. We did not compute the RMSE, NSE and DTW for the total SY and EC (see comment above), but solely for the area-averaged, i.e. one average value of EC (of the whole plot) per time step or the SY value up to that time step, data.

- Have you considered calibrating the model with the lab data and testing it against the field data or vice versa?
- No, we did not consider this. This is however a very nice idea and at the moment we are working at the calibration of plot data of a rainfall simulation in Saxony, Germany using time-lapse data to then use these models to apply them to hillslope data (as we also have time-lapse data of hillslopes; Grothum, O., Epple, L., Bienert, A., Eltner, A. (2024): Beobachtung und Rekonstruktion von Bodenerosionsprozessen mit permanenten Kamerastationen. Publikationen der DGPF e.V., Band 32, 106-115 *https://www.dgpf.de/src/tagung/jt2024/proceedings/band_32/dgpf2024_tagungsband_32.pdf*)

- Instead of searching for 'best models runs', have you considered looking at model realisations within the observational data's measurement errors? A 'best' model is always only 'conditionally-best' (e.g. on the range of conditions used for calibration, on imperfect evaluation data, prior assumptions, boundary conditions, and the criteria used for evaluating the model – as you have shown) (Beven, 2009, 2012). Using a limits-of-acceptability approach based on the errors in the observational data would allow you to identify behavioural model realisations while avoiding Type I errors (rejecting good models because of uncertain forcing data) (Beven, 2019, 2018). The behavioural realisations could still be analysed in light of the objective functions you defined, but without this quest for a single 'true' parameter set that would optimise all functions.
- This is an important point, which we however did not consider in this study as this would be beyond the scope of the current version. Nevertheless, this would be a next step to also consider the observation uncertainties and their error propagation into the objective functions to ideally also being able to provide a distribution of good parameters (including their covariances/correlations), which we highlighted at the end of our discussion in the manuscript.

Figure 4: The font size for the axis text in the DTW EC and DTW SY panels is too small. The legend for the rasters is missing – I do not think referring to the legend from figure 2 is very helpful here. Moreover, it would be nice to identify the panels (e.g. a, b, c…) to improve readability.
Thank you for the comment. We change the figure accordingly.

L339: Where is this shown in Figure 6?
Please, excuse this error. This should have been Figure 4. We correct the manuscript.

L345-347: This is very cool!
Thank you!

L349-350: Maybe the DEM smoothing is necessary for this kind of model application. Edit: Why was this not necessary for the lab rainfall simulation?
In regard of smooth DEM, we actually refer to smooth DEM changes, i.e. smooth DoDs. Smoothing the DEM as input does not necessarily avoid this circumstance as it is likely that still these artefacts due to very strong gradients of surface changes (e.g., strong erosion next to strong accumulation – but very locally) appear, just for larger cell sizes. It was not the case in the laboratory because here the erosion conditions are very different due to the significantly higher slope leading to fast development of rills and potentially giving less weight to the splash erosion aspect.

L355: I did not understand that "the metrics capture different aspects of soil surface change, including erosion". The example in the next sentence did not clarify your point to me. Moreover, are there any processes potentially leading to changes in the soil surface that RillGrow does not represent?

We are sorry for this misunderstanding. The sentence should have been "due to erosion" instead of "including erosion". To your last question, indeed there are other processes such as compaction or swelling and shrinking, which are not captured by RillGrow. In the discussion, we discuss this aspect and we refer to the study by Kaiser et al. (2018), which investigates this aspect in more detail. In our case study, the soil had a higher bulk density due the preparation to the make surface denser, thus no compaction was observed. And the shrinking/swelling was neither observed due to clay minerals being present that are not prone to these processes.

L359-363: This is what I meant above – a single model realisation that optimises all functions is irrelevant. If you choose different functions, repeat the rainfall simulation experiment, or change any steps in the DEM processing, you'll end up with a different optimal parameter set. Moreover, what is a good fit in this case? How do you define if the realisation fits a function "well"? I suggest rephrasing this to something along the lines of "We are looking to explore the behavioural parameter space constrained by different sources of data and objective functions".

We agree that the phrasing that we are searching for the one optimal set of model parameters is problematic. Overall, we are searching for a range of sets that best fit to our objective function assuming that we then find the best sets of parameters to predict erosion with RillGrow. We already see in the results that there is no such thing as that one best set of parameters. However, the overall approach remains that we use the objective functions to eventually optimize our costs, i.e. finding the best parameters by minimizing or maximizing the metric outputs. We rephrase the manuscript accordingly.

L364-365: This is a great demonstration of the equifinality problem!

Thank you for highlighting this point.

L378: Where are these metrics being used for "validation"? From what I understand, so far you have explored different metrics as part of the model calibration procedure.

We are sorry for the confusion. We change the word to evaluation.

L389: What does it mean that the model does not predict splash or interill erosion? Is this identified by a given parameterisation or by the outputs? Moreover, I thought RillGrow did not differentiate between rill and interill processes (L207).

This was identified by the final DoD, i.e., shown in figure 6. There, we can see that no erosion is predicted between the rills, which is not confirmed by the camera-based measurement (figure 2).

Regarding the second question, RillGrow does not distinguish between rill- and inter-rill flow erosion. Splash redistribution is calculated over the whole of the DEM, but is only effective where water depths are zero or shallow (i.e. in inter-rill areas).

L390: Do you mean more splash is modelled for the realisations with better DTW and SY metrics?

Yes, indeed. We change the manuscript.

You go into a lot of detail describing single model realisations, which makes the text long and sometimes difficult to follow. I think this stems from your focus on identifying a single

realisation to optimise all functions. I suggest focusing on more generalisable patterns and shortening some of the results.

We will shorten the results considering your suggestions.

L405-406: Calibration and validation seem to get confused, please check or define this somewhere. Beven (2009) defines calibration as "the process of adjusting parameter values of a model to obtain a better fit between observed and predicted variables". A calibrated model can then be tested against new data not used doing the calibration procedure (Klemeš, 1986). After reading the manuscript, I understand you tested different data and functions for calibrating RillGrow. Of course, this can be considered part of an evaluation process, but I suggest being precise about the terminology.

Thank you for highlighting this. We change all referrals to validation to calibration as we consider indeed no validation, yet. At best, we can consider our approach for calibration as we try to find the best set of parameters that lead to the least deviation between observation and model.

L412: Could this result from model input variables changing during the simulation and this not being picked up by the model parametrisation?

Yes, this might be a reason because input variables do not change during a RillGrow simulation.

L415: Please try to be more precise when describing model performance. What is a very close fit to the observation?

Our assessments in regard of model performance are at the moment more qualitative, e.g., by looking at the graphs to see how well they match. However, we added the numbers to the metrics (i.e., the lowest or highest to the $30^{th}$ values) of the best 30 sets of chosen parameters in figure 7 and figure 13 to be more quantitative in our assessment.

L440: Similar error metrics?

We change the sentence to the following "…good model fit according to the difference between observation and prediction described by the different calibration metrics."

Figure 9: Please add a legend for the point colours. Moreover, while the ten 'best' realisations can be very scattered, using a larger number of behavioural realisations might help you identify and describe patterns in the dotty plots.

We did not include a legend because no added value is given by it as each point color solely indicates which points belong to the same model run. Therefore, we would like to keep it this way. In regard of best model runs, we also checked higher numbers of realizations and thereby indeed the pattern changed a bit for the not so obvious relationships. However, the main findings remained. Nevertheless, we rephrase the manuscript to focus less strongly on this aspect as there might be more suitable approaches of joint consideration of the objective function outputs, which is beyond the scope of this study. In future studies more focus should be laid on how to best combine the different objective functions as our approach of sorting according to the best model runs might not be the most suitable as we discuss this in the discussion later on.

L453-455: Is this a limitation of the model or the data? As you mentioned above, the initial changes in the soil surface are too small to be detected, considering the DEM errors.

This is a limitation by the model because we refer here to the artefacts, which display very strong accumulation/erosion in very close proximity. After these artefacts are removed due to our filtering, mostly realizations remain, which use lower splash efficiency values.

Figure 10: Here the comma is used as a decimal separator.
Thank you for noticing. We corrected this.

L455-464: I had a hard time understanding this paragraph. What are these ranges? Which parameters do they represent?
We shorten this paragraph strongly to make the results more concise and focus on the most important aspects. Accordingly, also the figure and figure 15 are moved to the appendix. The original idea was that if we find the best model realizations that these would depict comparable parameters (or at least within a small range of deviation to each other only) and that we could find these model runs by jointly considering different objective functions (i.e., our ten choices) or at least some minimal combination of objective functions. However, our findings are not conclusive.

Figure 11: Would be great to have the observed DoD here. Also, shouldn't the abbreviations in the panel titles be described in the figure legend?
The observed DoD is provided in figure 2 and is not repeated here due to space constraints. We would like to avoid repeating the abbreviations again. However, we refer to the table where they are explained in the caption of the revised manuscript.

L530-535: It makes sense that the same model realisations that simulate higher changes in elevation also simulate higher sediment yield, right? The output variables should be correlated. Getting the rill patterns right is a different story.
Yes, this would be expected. But it is more obvious for the laboratory experiment (higher rill erosion) than the field experiment. This is what we aimed to portray here.

Figure 15: Please check the decimal separators. Why doesn't the y-axis start at zero? I also don't understand this figure; what is this parameter range?
Please, see our answer to figure 10 and Line 455-464.

L580-595: There have been multiple attempts to evaluate erosion models using spatial data, e.g. from field surveys, aerial images, and fallout-radionuclide data (Brazier et al., 2001; Fischer et al., 2018; Jetten et al., 2003; Saggau et al., 2022; Vigiak et al., 2006; Wilken et al., 2020). So, I am not sure that model evaluation has lagged behind the models – the technology is out there; the problem is that it is so much easier not to use it.
Thank you for making this point. We adapt our statement accordingly.

What I think is really unique and exciting in your approach is the quality, the spatiotemporal resolution, and the different sources of data (plot outlet and SfM) used for model calibration.
Thank you.

L605-610: Yes, I found this similarity index very useful!
Thank you.

References:

Beven, K.: Towards a methodology for testing models as hypotheses in the inexact sciences, Proc. R. Soc. A Math. Phys. Eng. Sci., 475(2224), doi:10.1098/rspa.2018.0862, 2019.

Beven, K. J.: Environmental Modelling: An Uncertain Future, Routledge, Oxon., 2009.

Beven, K. J.: Rainfall-Runoff Modelling, 2nd ed., John Wiley & Sons, Chichester., 2012.

Beven, K. J.: On hypothesis testing in hydrology: Why falsification of models is still a really good idea, WIREs Water, 5, e1278, doi:10.1002/wat2.1278, 2018.

Beven, K. J. and Young, P.: A guide to good practice in modeling semantics for authors and referees, Water Resour. Res., 49(8), 5092–5098, doi:10.1002/wrcr.20393, 2013.

Brazier, R. E., Beven, K. J., Anthony, S. G. and Rowan, J. S.: Implications of model uncertainty for the mapping of hillslope-scale soil erosion predictions, Earth Surf. Process. Landforms, 26, 1333–1352, 2001.

Cândido, B. M., Quinton, J. N., James, M. R., Silva, M. L. N., de Carvalho, T. S., de Lima, W., Beniaich, A. and Eltner, A.: High-resolution monitoring of diffuse (sheet or interrill) erosion using structure-from-motion, Geoderma, 375(May), 114477, doi:10.1016/j.geoderma.2020.114477, 2020.

Fischer, F. K., Kistler, M., Brandhuber, R., Maier, H., Treisch, M. and Auerswald, K.: Validation of official erosion modelling based on high-resolution radar rain data by aerial photo erosion classification, Earth Surf. Process. Landforms, 43(1), 187–194, doi:10.1002/esp.4216, 2018.

Jetten, V., Govers, G. and Hessel, R.: Erosion models: Quality of spatial predictions, Hydrol. Process., 17(5), 887–900, doi:10.1002/hyp.1168, 2003.

Kaiser, A., Erhardt, A., Eltner, A.: Addressing uncertainties in interpreting soil surface changes by multi-temporal high resolution topography data across scales. Land Degradation & Development, 29(8), 2264-2277, doi: 10.1002/ldr.2967, 2018

Klemeš, V.: Operational testing of hydrological simulation models, Hydrol. Sci. J., 31(1), 13–24, doi:10.1080/02626668609491024, 1986.

Kubínová, R., Neumann, M., Kavka, P.: Aggregate and Particle Size Distribution of the Soil Sediment Eroded on Steep Artificial Slopes. Appl. Sci, 11, 4427. doi: 10.3390/app11104427l, 2021.

Onnen, N., Eltner, A., Heckrath, G., Van Oost, K. (2020): Monitoring soil surface roughness under growing winter wheat with low altitude UAV sensing. Earth Surface Processes and Landforms, 45(14), 3747-3759

Oreskes, N.: Evaluation (not validation) of quantitative models, Environ. Health Perspect., 106(6), 1453–1460, doi:10.1289/ehp.98106s61453, 1998.

Oreskes, N., Shrader-Frechette, K. and Belitz, K.: Verification, validation, and confirmation of numerical models in the Earth Sciences, Science (80-. )., 263, 641–646, doi:10.1126/science.263.5147.641, 1994.

Saggau, P., Kuhwald, M., Hamer, W. B. and Duttmann, R.: Are compacted tramlines underestimated features in soil erosion modeling? A catchment-scale analysis using a process-based soil erosion model, L. Degrad. Dev., 33(3), 452–469, doi:10.1002/ldr.4161, 2022.

Takken, I., Beuselinck, L., Nachtergaele, J., Govers, G., Poesen, J. and Degraer, G.: Spatial evaluation of a physically-based distributed erosion model (LISEM), Catena, 37(3–4), 431–447, doi:10.1016/S0341-8162(99)00031-4, 1999.

Vigiak, O., Sterk, G., Romanowicz, R. J. and Beven, K. J.: A semi-empirical model to assess uncertainty of spatial patterns of erosion, Catena, 66(3), 198–210, doi:10.1016/j.catena.2006.01.004, 2006.

Warren, S. D., Mitasova, H., Hohmann, M. G., Landsberger, S., Iskander, F. Y., Ruzycki, T. S. and Senseman, G. M.: Validation of a 3-D enhancement of the Universal Soil Loss Equation for prediction of soil erosion and sediment deposition, Catena, 64(2–3), 281–296, doi:10.1016/j.catena.2005.08.010, 2005.

---

## Author Comment (AC2)

This is one of the most interesting and insightful studies I have read over the past few years. First, I would like to give a high respect to the heavy workload and computation efforts that the authors endeavored to produce such a meticulously designed and solid results. To be honest, it took me a couple of days to really finish reading through this excellent work. Very impressive.

As one of the most advanced techniques, cellular automation becomes increasingly popular in modeling soil erosion processes. In the Introduction part, the authors did a good job in reviewing and summarizing the state-of-the-art of cellular automation in the field of soil erosion modeling. This therefore makes it quite easy to comprehend the knowledge gaps in current studies, and in turn not difficult to understand the novelty of this study. Yet, I reserve some concerns that the authors may consider to include in the revised manuscript.

Thank you very much for your encouraging words and thank you very much for your comments and suggestions for change, which helps a lot to improve our manuscript. We are addressing each of your comments below. We are particularly grateful to you for taking two whole days in order to read thoroughly: thank you again!

- The opening of the Abstract seems a bit too long, and key research questions or knowledge gaps come in a bit too late. It would be better if the authors could specify the research question of this study more explicitly, and also save more writing space to present the key findings in the following part. Apart from the methods, the results and observations are a bit too general. More specific data or quantitative descriptions would be more helpful to underscore the novelty.

- Thank you for your suggestions. However, respectfully, we disagree regarding the opening of the Abstract. Given the current dominance of wholly empirical RUSLE-GIS approaches in soil erosion modelling, we submit that it is necessary to point out the "need for erosion models, necessarily process-focused, which are able to reliably represent rates and extents of soil erosion under unprecedented circumstances."
  Again, with respect, we disagree regarding the need to specify the research question of this study more explicitly. On line 15 of the abstract we state the research objective: „This study explores the use of Structure-from-Motion photogrammetry as a means to calibrate and evaluate this model". We suggest that this is sufficiently explicit. We have however inserted line breaks in the rather monolithic Abstract, to hopefully make it easier to spot the research objective during a quick scan of the Abstract.
  It is true that we wait until line 120, at the end of the Introduction, to state the research objective in the main body of the text: „The aims of this study were first to use several multi-objective functions to calibrate and evaluate a process-focused soil erosion model (RillGrow), and then to evaluate these objective functions in terms of information gained from each function." We chose to do this because it seems essential to us to summarise the context for this research before stating the research objective.
  Also, we are afraid that we are unclear regarding your suggestion that our results and observations "are a bit too general". Surely any research necessarily focuses on the specific to make inferences about the general? Put another way: we can't test everything, so we have to choose a test case and then extrapolate out from that.
  Finally, (and we again ask this with all due respect!) what exactly do you mean by "More specific data or quantitative descriptions would be more helpful to underscore the novelty"?

- In general, the Results part is a bit too long and saturated with figures. I would suggest remove some figures into the supplementary, or selectively show some of the subfigures. I believe, there must have been far more figures plotted out during the

entire modelling and analyzing. The authors must have already tried a lot to reduce the number of illustrations. Yet, still, as a piece of a regular research article, too many figures and too lengthy results somehow might make the readers feel overwhelmed. Is it possible to include a table listing out the major performances of different model runs under the three different approaches? Or, the authors may even consider to develop a sort of "graphical abstract" or "conceptual diagram" to summarize the research questions, critical methods and key findings? So as to help the authors develop a "holistic" comprehension over this study?

- Indeed, the result descriptions are long. However, we also performed various comparisons. However, we will reconsider each description of our findings to evaluate whether it could be shortened and/or moved to the supplement.
  Unfortunately, a table summarizing the results and describing the performance would also be too long and would not bring more clarity. For follow-up work, we would like to refer to our provided raw data including python scripts to run the analysis (including a short description on how to use them) and thus being able to have an even more detailed insight.
  The graphical abstract / conceptual diagram is a great idea and will be provided.

- In addition, there are always bits and pieces of discussion mixed in the Result part. This on the one hand makes the Results part quite lengthy; on the other hand, in the current state, the Discussion part is more into "limitations and future implications", but short in in-depths explanations, coherent arguments and discussions with other peer studies (most of which is actually scattering in Results).
- We check the manuscript and revise it accordingly to make the results and discussion more coherent.

- L580 to L595 in the Discussion section is actually a review over currently available models. They should be moved to the Introduction part, to better specify the knowledge gaps of current studies.
- We move the section accordingly.

- Some of the results were described in present tense. Should they be in past tense? For instance, L485 to L515 in subsection 3.2.
- Thank you for noticing. We will revise this.

- Although the authors mentioned the specific subprocesses, such as raindrop detachment, splash transport, flow transport and flow detachment (mostly derived from Kinnell 2001), the potential effects or impacts of these subprocesses were not adequately discussed. For instance, the selectivity of runoff over eroding time in carrying soil particles of different sizes. This may partly contribute to the unmatched temporal variations of sediment yield.
- Thanks! This is indeed something that we wished to focus on more when writing the paper. However, the paper is already rather long (as you yourself point out). Lengthening the paper to include this extra discussion would not be a good idea. Also, we have learned a good deal from this work and, as a result, the RillGrow model is now being modified to better represent some of these erosional subprocesses (splash redistribution in particular). Therefore we suggest that discussion of the effects of strengths and weaknesses in RillGrow's representation of these subprocesses, and their implications, would be better kept back for a follow-up paper.
- Furthermore, this paper focuses primarily on modelling the development of rills. During the simulations, the grain composition of the sediment was measured and not

only the grain composition but also the disintegration (by ultrasound) of these particles was evaluated (methodology described in Kubinova et al., 2021). Based on our results so far, which have been carried out on steep slopes where erosion rills have formed, the grain size distribution of the eroded soil does not differ significantly from that of the original soil throughout the simulation. The generally accepted concept of selective erosion was not confirmed in these experiments, and the use of a model could help to refine erosion processes in the future. However, this "sub-topic" is beyond the scope of this manuscript.

- L615, the statement on "erosion model calibration might use EC-based measures only, and even possible without using sediment yield" is somewhat a bit bold, I think. That the relatively smaller errors of EC-based approaches were valid in this study, at least to some extent, was because the soil surface was prepared with compaction and heavy bulk density. Some minor changes in EC, such as the settling of soil surface after wetting, the removal and in turn runout of depositional sediment over time, the periodic initiations of different rills and rejuvenated eroding surface, and the progressive equilibrium of runoff and sediment in the intervals of rill development, may trigger major changes in sediment yield.
- Indeed, the removal and in turn runout of depositional sediment over time, the periodic initiations of different rills and rejuvenated eroding surface, and the progressive equilibrium of runoff and sediment in the intervals of rill development, can trigger major changes in sediment yield, which however should be measurable by the DoDs as these processes result in direct changes of the soil surface height. However, with the settling, this is a different picture. This needs to be considered differently. Here, we already investigate, how approximations of settling processes might help to disentangle this process from erosion or at least indicate when both processes are happening at the same time (Epple et al., moderate revision, Soil and Tillage Research). In this case sediment yield measurement is needed if indeed erosion is to be measured. We made our statement less bold in the revised manuscript.

Overall, this is a well-written manuscript offering a huge amount of information and new thoughts. Yet, I think it would be even better appreciated, if the authors may consider to trim and condense it a bit shorter (even just for the sake of less APC 😊).
Thank you for your comment. We shorten the manuscript accordingly.

References:

Beven, K.: Towards a methodology for testing models as hypotheses in the inexact sciences, Proc. R. Soc. A Math. Phys. Eng. Sci., 475(2224), doi:10.1098/rspa.2018.0862, 2019.

Beven, K. J.: Environmental Modelling: An Uncertain Future, Routledge, Oxon., 2009.

Beven, K. J.: Rainfall-Runoff Modelling, 2nd ed., John Wiley & Sons, Chichester., 2012.

Beven, K. J.: On hypothesis testing in hydrology: Why falsification of models is still a really good idea, WIREs Water, 5, e1278, doi:10.1002/wat2.1278, 2018.

Beven, K. J. and Young, P.: A guide to good practice in modeling semantics for authors and referees, Water Resour. Res., 49(8), 5092–5098, doi:10.1002/wrcr.20393, 2013.

Brazier, R. E., Beven, K. J., Anthony, S. G. and Rowan, J. S.: Implications of model uncertainty for the mapping of hillslope-scale soil erosion predictions, Earth Surf. Process. Landforms, 26, 1333–1352, 2001.

Cândido, B. M., Quinton, J. N., James, M. R., Silva, M. L. N., de Carvalho, T. S., de Lima, W., Beniaich, A. and Eltner, A.: High-resolution monitoring of diffuse (sheet or interrill) erosion using structure-from-motion, Geoderma, 375(May), 114477, doi:10.1016/j.geoderma.2020.114477, 2020.

Fischer, F. K., Kistler, M., Brandhuber, R., Maier, H., Treisch, M. and Auerswald, K.: Validation of official erosion modelling based on high-resolution radar rain data by aerial photo erosion classification, Earth Surf. Process. Landforms, 43(1), 187–194, doi:10.1002/esp.4216, 2018.

Jetten, V., Govers, G. and Hessel, R.: Erosion models: Quality of spatial predictions, Hydrol. Process., 17(5), 887–900, doi:10.1002/hyp.1168, 2003.

Kaiser, A., Erhardt, A., Eltner, A.: Addressing uncertainties in interpreting soil surface changes by multi-temporal high resolution topography data across scales. Land Degradation & Development, 29(8), 2264-2277, doi: 10.1002/ldr.2967, 2018

Klemeš, V.: Operational testing of hydrological simulation models, Hydrol. Sci. J., 31(1), 13–24, doi:10.1080/02626668609491024, 1986.

Kubínová, R., Neumann, M., Kavka, P.: Aggregate and Particle Size Distribution of the Soil Sediment Eroded on Steep Artificial Slopes. Appl. Sci, 11, 4427. doi: 10.3390/app11104427l, 2021.

Onnen, N., Eltner, A., Heckrath, G., Van Oost, K. (2020): Monitoring soil surface roughness under growing winter wheat with low altitude UAV sensing. Earth Surface Processes and Landforms, 45(14), 3747-3759

Oreskes, N.: Evaluation (not validation) of quantitative models, Environ. Health Perspect., 106(6), 1453–1460, doi:10.1289/ehp.98106s61453, 1998.

Oreskes, N., Shrader-Frechette, K. and Belitz, K.: Verification, validation, and confirmation of numerical models in the Earth Sciences, Science (80-. )., 263, 641–646, doi:10.1126/science.263.5147.641, 1994.

Saggau, P., Kuhwald, M., Hamer, W. B. and Duttmann, R.: Are compacted tramlines underestimated features in soil erosion modeling? A catchment-scale analysis using a process-based soil erosion model, L. Degrad. Dev., 33(3), 452–469, doi:10.1002/ldr.4161, 2022.

Takken, I., Beuselinck, L., Nachtergaele, J., Govers, G., Poesen, J. and Degraer, G.: Spatial evaluation of a physically-based distributed erosion model (LISEM), Catena, 37(3–4), 431–447, doi:10.1016/S0341-8162(99)00031-4, 1999.

Vigiak, O., Sterk, G., Romanowicz, R. J. and Beven, K. J.: A semi-empirical model to assess uncertainty of spatial patterns of erosion, Catena, 66(3), 198–210, doi:10.1016/j.catena.2006.01.004, 2006.

Warren, S. D., Mitasova, H., Hohmann, M. G., Landsberger, S., Iskander, F. Y., Ruzycki, T. S. and Senseman, G. M.: Validation of a 3-D enhancement of the Universal Soil Loss Equation for prediction of soil erosion and sediment deposition, Catena, 64(2–3), 281–296, doi:10.1016/j.catena.2005.08.010, 2005.

---

## Author Response (AR2)

Dear Pedro, thank you very much for providing the first and second review. It helped very much to improve the manuscript.

The authors replied to all comments from the other referee and me. The changes in the manuscript have made it more precise and easier to digest.
Some of the figures could still use some attention. In Figure 9, I see the colours in the dotty plots but do not know what they mean. At the very least, this must be stated in the figure caption.
Thank you for the comment. We added some explanation to the caption.

Figure 9 and many others are pixelated, so I cannot read the axis labels and legends.
The figures are pixelated because they are only provided in the pdf directly (converted from Word). I replaced them with high resolution images, but during the pdf conversion the quality is reduced again. However, for the final version the original figures should be used.
@Editor: Can you please verify if the original image files are requested for the typesetting?

Moreover, Figures 10 and 15 from the previous version are still in the manuscript and the ATC files, although I suppose the authors wanted to remove these figures.
Thank you for noticing the important error on page 20 and 24. Indeed, these figures should not have been there. A struggle with Word…

In Figure 11, the abbreviations/labels on each panel are confusing. Why not use a letter to identify each panel and then give the details in the figure caption?
Thank you for the suggestion. However, we prefer it this way to keep the figure caption short and we refer to table 2 where all abbreviations are explained.

This is such a nice manuscript, and I would like to see it presented better.
Best wishes,
Pedro